# Evaluation of hydroclimatic biases in the Community Earth System Model (CESM1) within the Mississippi River basin

Michelle O'Donnell[1], Kelsey Murphy[2], James Doss-Gollin[3], Sylvia Dee[2], Samuel Munoz[1,4]

[1]Department of Civil & Environmental Engineering, Northeastern University, Boston, MA, USA
[2]Department of Earth & Planetary Sciences, Rice University, Houston, TX, USA
[3]Department of Civil & Environmental Engineering, Rice University, Houston, TX, USA; [4]Department of Marine & Environmental Sciences, Marine Science Center, Northeastern University, Nahant, MA, USA

*Correspondence to*: Michelle O'Donnell (odonnell.miche@northeastern.edu)

**Abstract**. The Mississippi River is a critical waterway in the United States, and hydrologic variability along its course represents
a perennial consideration for trade, agriculture, industry, ecosystems, the economy, and communities. Simulations of past, historic, and projected river discharge have been widely used to assess the dynamics and causes of changes in the hydrology of the Mississippi River basin over long time scales and to put changes of climate in context. The Community Earth System Model version 1 (CESM1) offers such simulations to complement observational records of river discharge by providing fully coupled output from a state-of-the-art earth system model that includes a river transport model. Here, we compare observations and
reanalysis datasets of key hydrologic variables to CESM1 output within the Mississippi River basin to evaluate model performance and bias. We show that the seasonality of simulated river discharge in CESM1 is shifted 3 months late relative to observations. This offset is attributed to seasonal biases in precipitation and runoff in the region. We also evaluate performance of several Coupled Model Intercomparison Phase 6 (CMIP6) models over the Mississippi River basin, and show that runoff in other models — notably the Community Earth System Model version 2 (CESM2) — more closely simulates the seasonal trends in the reanalysis
data. Our results have implications for model selection when assessing hydroclimate variability on the Mississippi River basin, and show that the seasonal timing of runoff can vary widely between models. Our findings point to 1) a need for continued developments in the representation of land surface hydrology in earth system models for improvements in our ability to assess the causes and consequences of environmental change on terrestrial water resources and major river systems globally, and 2) a need for caution and understanding of biases when applying these tools to practical risk assessment.

## 1 Introduction

Ongoing and projected changes in streamflow due to climate change remain uncertain because of the complex and dynamic nature of river systems and the interactions between the ocean, atmosphere, and land surface that govern terrestrial hydrologic processes (Clark et al., 2015; Fisher and Koven, 2020; Good et al., 2015; Wood et al., 2011). Our understanding of hydrologic changes is informed by observational datasets, but earth system and hydrologic models play an increasingly critical role in
examining the impacts of climate variability and climate change on river discharge as systems vary outside of what has previously been observed as normal (Fowler et al., 2022; Herrera et al., 2023; Milly et al., 2008). However, several key hydrologic processes that regulate river discharge remain poorly constrained in earth system models. This results in uncertainties around future streamflow conditions that represent a critical challenge for water resources management, hazard mitigation, and emergency response (Fowler et al., 2022; Her et al., 2019; Troin et al., 2022; Vetter et al., 2017). While understanding changes in
river discharge and its repercussions for management is important across multiple spatial and temporal scales, it is particularly important for large river systems, like the Mississippi River basin (Figure 1), which serve as regional economic arteries for hydroelectric power, transportation, and fresh water for municipal, industrial, and agricultural use.

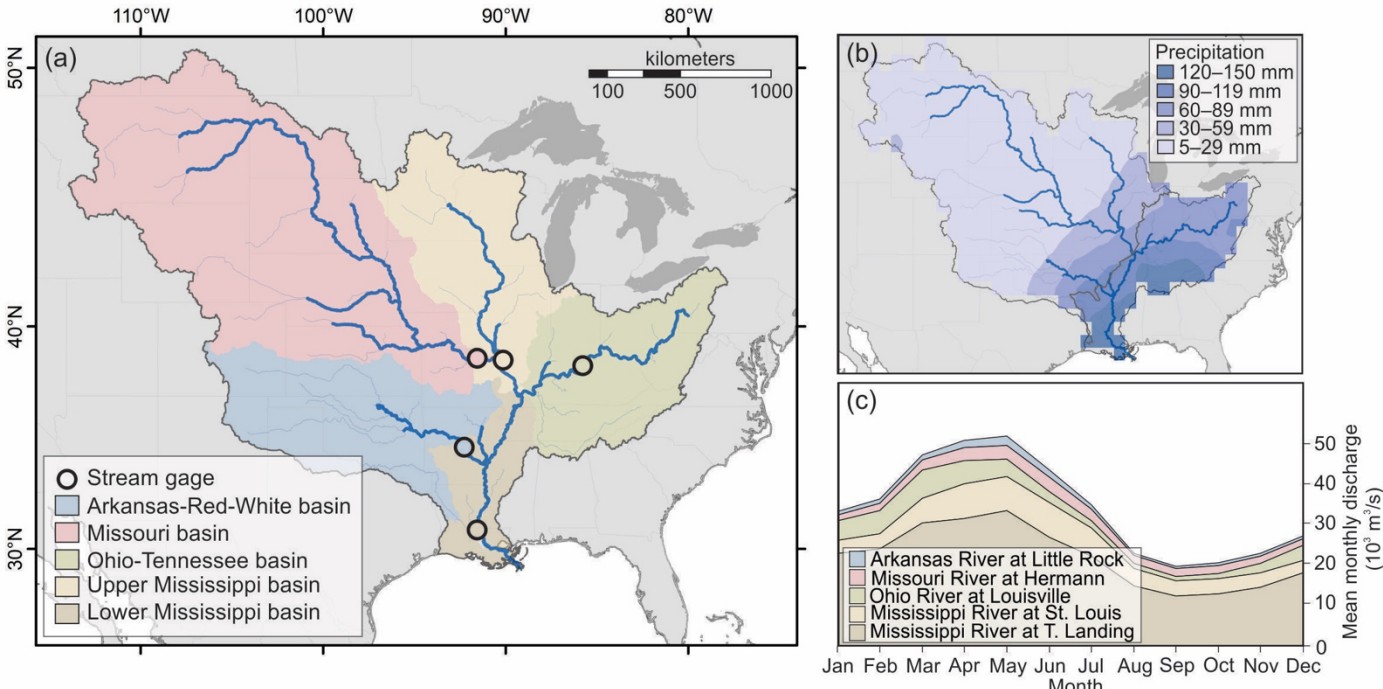

Figure 1. Mississippi River Basin and major tributaries: (a) Basins of major tributaries and corresponding stream gage locations. (b) Monthly mean (mm/month) precipitation from Global Precipitation Climatology Center (GPCC)[10] and grouping of subbasins into Eastern and Western Basins (gray line). (c) Monthly mean discharge from stream gages on the major tributaries (1950–2010).

How the Mississippi River is responding, and will continue to respond to, changes in climate and land use is uncertain, exemplifying the unknowns inherent to many of the world's large temperate river systems (Fowler et al., 2022). At present, it is unclear whether recent changes in Mississippi River streamflow should be attributed primarily to changes in climate or to human modifications to the land surface and river channel (Criss and Shock, 2001; Munoz et al., 2018; Pinter et al., 2008; Watson et al., 2013). Precipitation over the upper Mississippi River basin has increased by 0.66 mm per year (Ziegler et al., 2005) but evapotranspiration has also increased since the late 20th century (Mccabe and Wolock, 2019; Qian et al., 2007). Observations alone cannot fully constrain these changes, as monitoring networks can be sparse, inconsistent, or have data that is difficult to access depending on the hydrologic variable (Fekete and Vörösmarty, 2007).

Global climate models, particularly those with ensemble runs, offer one way to explore the causes of historic hydrologic changes, and possible changes in projected hydrologic conditions. However, projections of streamflow remain uncertain, with modeling studies documenting both increases in river discharge (Tao et al., 2014) and decreases in Mississippi river discharge (van der Wiel et al., 2018) over the 21st century in response to climate change. The disparities in these streamflow projections reflect, in part, the use of different models and emissions scenarios, related hydrologic parameters remaining difficult to constrain, and the challenges of validating models against observations on a river system that has been heavily modified by human activities. Some uncertainty can be constrained by the use of multiple models or model ensembles as they are run into the future for different scenarios (Thackeray et al., 2022; Velázquez et al., 2011). At the same time, artificial reservoirs, levees, cutoffs, and spillways constructed primarily during the mid-20th century remain challenging to incorporate into hydrologic

models (Brookfield et al., 2023; Tavakoly et al., 2021). Additionally, it is not standard for river routing to be incorporated into earth system models.

One approach to evaluate the roles of climate variability and change on streamflow is to use a fully coupled earth system model that includes a hydrologic model; one such model widely used for this purpose is the Community Earth System Model (CESM1). CESM1 is the only model among Coupled Model Intercomparison Project Phase 5 (CMIP5) and Phase 6 (CMIP6) models that has both a river routing model and a Last Millennium Ensemble (CESM-LME) project (Otto-Bliesner et al, 2016). This includes both full-forcing and single-forcing simulations for the period 850-2005 CE. In addition to simulating hydrologic processes included in all CMIP models (i.e., precipitation, soil moisture, runoff), CESM1 includes a River Transport Model (RTM) that simulates river discharge at daily and sub-daily time-steps on a finer resolution (0.25° grid). The RTM is included in multiple CESM1 experiments, including the LME. To date, there are no other projects that include equivalents of the LME project simulations along with simulated river discharge. Additionally, CESM1 has been widely used to examine the roles of climate variability, climate change, impacts of land cover changes on streamflow, and hydroclimate over the last millennium (Abram et al., 2020; Chen et al., 2024; Cresswell-Clay et al., 2022; Falster et al., 2023; Munoz and Dee, 2017; Murphy et al., 2024; Thapa and Stevenson, 2024; Wiman et al., 2021; Zhao et al., 2020). Despite the large potential for this particular hydrologic model coupled to an earth system model to study and resolve uncertainties in the response of streamflow to climate change, we currently lack a robust validation of the CESM1's terrestrial hydrology over a major temperate river basin, including the Mississippi River Basin.

Here, we validate output from CESM1 over the Mississippi River basin through comparisons to observed river discharge and climate reanalysis of other key hydrologic variables, including precipitation, soil moisture, snowmelt, evapotranspiration, and runoff. Specifically, we use monthly output from the Last Millennium Ensemble (LME) of CESM1, which provides 13 fully-forced ensemble members over the historic period, and compare simulated seasonal trends in all major hydrologic variables over multiple parts of the Mississippi River basin to stream gage observations (US Army Corps of Engineers, 2025; U.S. Geological Survey, 2016a, p.07, b, c, d, e) and fifth generation of European ReAnalysis (ERA5) (Muñoz-Sabater et al., 2021) from the 20[th] century to present. We show that, on all major tributaries of the Mississippi River, the seasonality of peak discharge in CESM1 is 3 months late relative to observations. We then show that the shifted seasonality of simulated river discharge is primarily due to an offset in the seasonality of simulated precipitation in CESM1, particularly over the eastern portion of the Mississippi River basin. Finally, we evaluate how Mississippi River basin hydrology in CESM1 compares to other CMIP6 models, and show that other models — notably CESM2, which also simulates river discharge, but with the updated hydrologic model Model for Scale Adaptive River Transport (MOSART) — are more skillful in simulating the observed seasonality of runoff. We conclude that recent improvements in earth system models with robust representations of terrestrial hydrology, specifically their simulations of runoff, represent an important step towards improving projections of water resources in the face of ongoing climate change.

## 2 Methods and Data

### 2.1 Subbasin Hydroclimate

The Mississippi River Basin spans a range of hydroclimatic conditions. The western portion of the basin, including the Arkansas, Missouri, and Upper Mississippi basins, receives an average monthly precipitation of 5–59 mm. The eastern portion of the basin, including the Ohio-Tennessee and Lower Mississippi, receives 60–150 mm/month. The subbasins are divided along the median range of precipitation values (~60 mm/month), which most closely follows a set of sub-basin boundaries (Figure 1b). The Entire

Mississippi basin can also be categorized into subbasins by other hydroclimate variables, including temperature, actual evapotranspiration, and runoff; when categorized by these variables, similar subbasin groupings emerge to those produced by

precipitation patterns (Mccabe and Wolock, 2019). While the basin can be divided and grouped at different scales, we refer to it as the Eastern Mississippi basin (Ohio-Tennessee and Lower Mississippi), Western Mississippi basin (Arkansas, Missouri, and Upper Mississippi basins) and entire Mississippi basin in subsequent sections of the discussion given similar precipitation and hydroclimate characteristics (Figure 1).

**2.2 Stream gage observations**

To evaluate the skill of CESM1 to simulate river discharge on the major tributaries of the Mississippi River basin, we first selected United States Geological Survey (USGS) and United States Army Corps of Engineers (USACE) streamflow gages from the lowermost reaches from the Upper Mississippi River, Missouri River, Ohio River, Arkansas River, and Lower Mississippi River. Gages were selected based on their geographic location as far downstream on the tributary and near to the confluence with the main stem of the Mississippi as possible, and for the length and continuity of daily streamflow data available; selected gages

include the Mississippi River at St. Louis, MO (07010000) (U.S. Geological Survey, 2016a), Missouri River at Hermann, MO (06934500) (U.S. Geological Survey, 2016c), Ohio River at Louisville, KY (03294500) (U.S. Geological Survey, 2016b), Arkansas River at Little Rock, AR (07263500) (U.S. Geological Survey, 2016d), and Mississippi River at Vicksburg, MS (07289000) (U.S. Geological Survey, 2016e) and Mississippi River at Tarbert Landing (01100Q) (US Army Corps of Engineers, 2025) (Table 1).  Both the Mississippi River at Vicksburg, MS and Mississippi River at Tarbert Landing are included to account

for the difference in flow volumes and periods of record.

**Table 1. Gage Statistics for gages used in analysis, including USGS gages Mississippi River at St. Louis, MO, Missouri River at Hermann, MO, Ohio River at Louisville, KY, Arkansas River at Little Rock, AR, and Mississippi River at Vicksburg, MS** (U.S. Geological Survey, 2016c, c, b, d, e) and **USACE gage Mississippi River at Tarbert Landing**, (US

Army Corps of Engineers, 2025).

| Tributary | Gage Name | Gage Number | Agency | Period of Record (Monthly Statistics) | Start Year | End Year | Length of Record (Years) |
|---|---|---|---|---|---|---|---|
| Upper Mississippi | Mississippi River at St. Louis, MO | 07010000 | USGS | 1861-01 to 2024-10 | 1861 | 2024 | 163 |
| Missouri | Missouri River at Hermann, MO | 06934500 | USGS | 1928-10 to 2024-10 | 1928 | 2024 | 96 |
| Ohio | Ohio River at Louisville, KY | 03294500 | USGS | 1928-01 to 2024-10 | 1928 | 2024 | 96 |
| Arkansas | Arkansas River at Little Rock, AR | 07263500 | USGS | 1927-10 to 1970-09 | 1928 | 1970 | 42 |
| Lower Mississippi | Mississippi River at Tarbert Landing | 01100Q | USACE | 1930-01 to 2024-10 | 1930 | 2024 | 94 |
| Lower | Mississippi River at | 07289000 | USGS | 2008-01 to 2022-09 | 2008 | 2022 | 14 |

| Mississippi | Vicksburg, MS | | | | | | |
|---|---|---|---|---|---|---|---|

From the daily streamflow data, we computed monthly means for the period of USGS or USACE record (Table 1) that overlaps with CESM1 data (850-2005). To evaluate the influence of human modifications on river discharge seasonality, we also

computed monthly means to compare flows for the period prior to and after the implementation of most artificial reservoirs and spillways (Table 2, Figure A1). River modifications were developed over multiple decades on the tributaries of the Mississippi River, and became operational at different times (Table 2). Overall, previous studies have identified periods of major modifications and shown that dams for navigation have no substantial impact on flood discharges, rivers tend to maintain intra-annual variability, volumes of flow may be altered, but overall the impacts of alterations decrease downstream of modifications

as smaller tributaries enter (Alexander et al., 2012; Jacobson and Galat, 2008; Keown et al., 1986; Remo et al., 2018). In observed gage data here, modifications impact discharge volumes, but the month of peak flow is the same for gages on the Lower Mississippi, Arkansas, Ohio, and Missouri Rivers, and only one month earlier for the Upper Mississippi after the end of major modifications. We discuss this further in Section 3.1: Seasonality.

**Table 2. Discharge statistics from USGS and USACE gages for pre- and post- river modifications, where modification dates are based on the end of major river engineering on the tributary** (Alexander et al., 2012; Jacobson and Galat, 2008; Keown et al., 1986; Remo et al., 2018)**. USGS gages include Mississippi River at St. Louis, MO, Missouri River at Hermann, MO, Ohio River at Louisville, KY, Arkansas River at Little Rock, AR, and Mississippi River at Vicksburg, MS** (U.S. Geological Survey, 2016c, c, b, d, e) **and USACE Mississippi River at Tarburt Landing** (US Army Corps of

Engineers, 2025).

| Tributary | Gage Number | Year of end of major modification | Pre-modification | | | | Post-modification | | | |
|---|---|---|---|---|---|---|---|---|---|---|
| | | | Month of peak flow | Mean Flow (cfs) | Max Flow (cfs) | Min Flow (cfs) | Month of peak flow | Mean Flow (cfs) | Max Flow (cfs) | Min Flow (cfs) |
| Upper Mississippi | 07010000 | 1980 | May | 113469 | 595806 | 4377 | April | 130112 | 474143 | 11336 |
| Missouri | 06934500 | 1967 | June | 69331 | 445226 | 6827 | June | 94308 | 376290 | 21558 |
| Ohio | 03294500 | 1975 | March | 113469 | 595806 | 4377 | March | 130112 | 474143 | 11336 |
| Arkansas | 07263500 | 1970 | May | 39848 | 290268 | 1141 | May | 39461 | 99987 | 8291 |
| Lower Mississippi | 01100Q | 1980 | April | 458796 | 1520000 | 75000 | April | 548065 | 1619000 | 111000 |
| Lower Mississippi | 07289000 | 1980 | na | na | na | na | May | 770456 | 1996909 | 217345 |

## 2.3 Reanalysis and gridded observations

To evaluate the hydrologic processes that contribute to Mississippi River discharge, and for validation of the CESM1 simulations, we use ERA5 (Muñoz-Sabater et al., 2021) and gridded observations of precipitation from the Global Precipitation
Climatology Center (GPCC) (Becker et al., 2013). From the ERA5 reanalysis, we use 2m temperature (t2m), Snowmelt (smlt), Runoff (ro), Surface runoff (sro), and Sub-surface runoff (ssro). From the Livneh hydrometeorological dataset (Livneh et al., 2013), we use total evapotranspiration (et). Finally, from the GPCC (Becker et al., 2013) dataset, we use precipitation (precip). Periods of data used were selected based on the earliest starting date and latest ending date common to each dataset and CESM1, respectively (Table 1). We use monthly means for GPCC, Livneh, and all ERA5 variables. Datasets were cropped to the extent
of the grouped subbasins (Eastern and Western), as well as to the entire extent of the Mississippi River Basin. Grid cells falling within each were averaged over each subbasin.

## 2.4 Earth system models and validation approach

CESM1 variables examined include river discharge (QCHANR), total precipitation (PRECC + PRECL to represent total precipitation; convective precipitation rate (liquid + ice) + large scale (stable) precipitation rate (liquid + ice)), evapotranspiration
(QSOIL + QVEG + QVEGT to represent total evapotranspiration), total liquid runoff (QRUNOFF), surface runoff (QOVER), subsurface runoff (QDRAI), temperature (TREFHT), and snow melt (QSNOMELT). Surface runoff at glaciers (liquid only), wetland, lakes (QRGWL), the remaining runoff variable available, was not included in the analysis of runoff terms since the Mississippi River Basin is not a glaciated area.

All CESM1 variables are from the CESM1 Last Millennium Ensemble Project (LME). Part of the CMIP5 suite of models, these model data are still widely used to investigate climate variability over the last millennium. Ensemble members included here are from the 13 simulations from 850-2005 CE with transient forcings where the ensemble spread is due to the application of a random roundoff in air temperature at the start of each run. Forcings for these runs include orbital, solar, volcanic, land use/land cover change, and greenhouse gas concentrations (Otto-Bliesner et al., 2016).


    CESM1 is the only CMIP model to include both a Last Millennium Ensemble project and modeled discharge. The land model used in CESM1 is the Community Land Model (CLM4.0), which incorporates the carbon cycle, vegetation dynamics, and river routing. The River Transport Model (RTM) routes total runoff from the CLM to the oceans and seas as discharge, closing the hydrologic cycle in the model. Discharge is calculated as:

$$Q_{out} = S\left(\frac{v}{d}\right),\tag{1}$$

where S is the storage of river water within the cell ($m^3$), v is the effective channel velocity and is a global constant, (0.35 m s-1), and d is the distance from the center of the current cell to the downstream cell. Travel time for subgrid routing is not incorporated. This calculation of runoff generation and river routing does not explicitly incorporate water management (Dai and Trenberth, 2002; Li et al., 2013; Otto-Bliesner et al., 2016). While CESM1 documentation states that RTM output for river flow

can be directly compared to gauging station data, CLM has been updated over time to improve deficiencies in simulations of the hydrologic cycle (CESM Overview, 2024; Oleson et al., 2010).

    To assess the skill of other models in the basin, data from six CMIP6 models was also compared to ERA5 and CESM1 runoff data for the major tributaries. CMIP6 model selection was guided by their previous application in other hydroclimate studies. Models were chosen if they had been used in studies at a major basin scale or larger, compared to other models, or used in

studies related to hydroclimate changes in North America (Dai and Nie, 2022; Feng et al., 2022; Ji et al., 2024; Yazdandoost et al., 2021). Models selected include BCC CSM2 MR, CanESM5, CESM2 FV2, MIROC6, MPI ESM1 2 LR, and MRI ESM2.0. While CMIP6 models have many common output variables, the majority do not include simulated river discharge, so only total runoff (mrro) is compared between models here.

    The ensemble mean of the 13 individual ensemble members of the CESM1 full forcings runs was calculated for each variable

being used for comparison (river discharge, total precipitation, evapotranspiration, total liquid runoff, surface runoff, subsurface runoff, temperature, and snowmelt). CESM1 river discharge was first compared to observed discharge using the grid cell corresponding to the corresponding USGS and USACE gages. All remaining datasets were cropped to the extent of the major subbasins (Upper Mississippi, Lower Mississippi, Ohio/Tennessee, Arkansas, Missouri), the grouped subbasins (Eastern and Western), as well as to the entire extent of the Mississippi River Basin. Grid cells falling within each were averaged over each

subbasin. For each variable, the monthly mean value is then plotted for CESM1 along with the corresponding reanalysis data.

    To assess the skill of the CESM1 model data, two primary metrics are used: lagged correlation and spectral angle. Both are calculated in Python with the pandas corr and HydroErr sa functions, respectively (Jackson et al., 2019; The pandas development team, 2024). Lag correlation is used to assess the timing of peak flow in each dataset, and if the peak is offset between datasets, what the optimal offset is. Spectral angle is useful in this context because it indicates how well the shape of two data series match

independently of differences in magnitude (Jackson et al., 2019). Relative difference is also calculated between simulated and observed or reanalysis data to assess the differences in magnitudes between datasets as:

$$\frac{(modeled-observed)}{observed} \times 100 \; , \tag{2}$$

where observed values are either observed values, or reanalysis values (Jackson et al., 2019; Michalek et al., 2023). Since the

focus here is on seasonality, lag correlation and spectral angle are more representative in understanding the causes in shifted seasonal timing of discharge in CESM1.

CESM1 and observed monthly discharge data are plotted and compared to establish the discrepancy in seasonality between the observed streamflow in the major tributaries of the Mississippi and the model output. Each hydrologic variable from CESM1 is compared to reanalysis data for general fit, then quantitatively assessed with the skill metrics of lag correlation and spectral

angle.

## 3 Results & Discussion

### 3.1 Simulated discharge and stream gage observations

Simulated river discharge in CESM1 exhibits biases in both the magnitudes and seasonality of observations relative to stream

gages (Figure 2) (Table 3). The timing of modeled discharges are delayed on all major tributaries relative to observations; in this section, we diagnose potential model biases contributing to this shift.

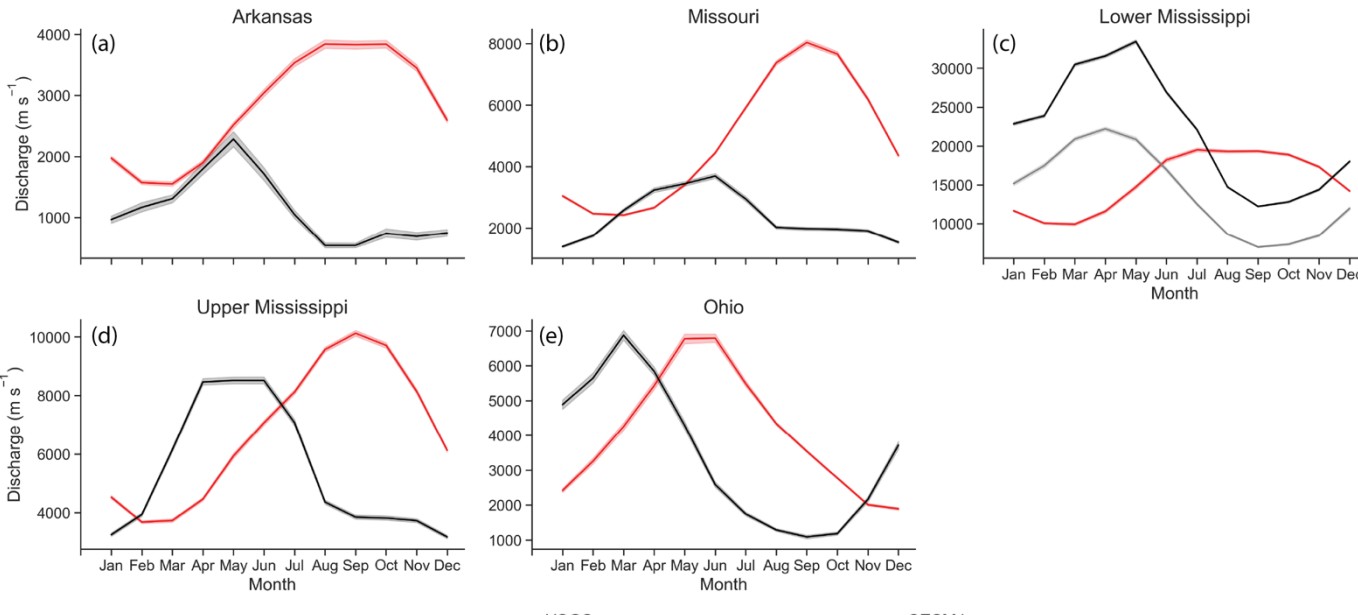

**Figure 2. Monthly mean CESM1 simulated river discharge (red) compared to observations from USGS (black) and USACE (gray) stream gages (black): (a) Arkansas-Red-White (Arkansas River at Little Rock, AR, 07263500), (b)**

**Missouri (Missouri River at Hermann, MO, 06934500), (c) Lower Mississippi (Mississippi River at Tarbert Landing, 01100Q and Mississippi River at Vicksburg, ), (d) Upper Mississippi (Mississippi River at St. Louis, MO, 07010000), (e) Ohio-Tennessee ( Ohio River at Louisville, KY, 03294500). Shading represents a 95% confidence interval from interannual variability.**

*Peak Annual Discharge.* CESM1 simulated annual peak (maximum) discharge is delayed relative to observations for all major tributaries. For the Missouri, Arkansas, and Upper Mississippi the magnitude of peak discharge is 18–117% too large, while the

Ohio and Lower Mississippi have simulated peak discharge values that are 1.22 and 12.13% smaller than the gage observations, respectively (Table 3). The major tributaries, including the Upper Mississippi, Missouri, Ohio, Arkansas-Red-White, Ohio Tennessee and Lower Mississippi, show a delay of three months in the timing of their peak discharge when CESM1 modeled

data is compared to USGS gage data. This means that simulated peak flows are occurring in June through September, with high flows extending into the fall, instead of aligning with observed USGS peak flows that occur from March into June.

**Table 3. Timing offset and relative difference values for hydroclimate variables between simulated (CESM1) and observed data (USGS or USACE for discharge) for maximum and minimum values. Timing offset is in months, where**
**positive values indicate simulated values are delayed relative to observations, and negative values indicate simulated values are early relative observations. Relative difference values are a percent, and positive values indicate that simulated values are larger, while negative values indicate that simulated values are smaller than observed values.**

| Variable | Basin | Maximum | | Minimum | |
|---|---|---|---|---|---|
| | | Timing offset (months) | Relative Difference (%) | Timing offset (months) | Relative Difference (%) |
| Discharge | Missouri | 3 | 117.62 | 2 | 74.74 |
| Discharge | Arkansas | 3 | 68.04 | -5 | 187.55 |
| Discharge | Ohio | 3 | -1.22 | 3 | 73.73 |
| Discharge | Upper Mississippi | 3 | 18.84 | 2 | 15.84 |
| Discharge | Lower Mississippi | 3 | -12.13 | -6 | 42.71 |

*Low-Flows.* CESM1 simulated low flows (annual minima) are delayed 2–3 months relative to USGS or USACE gage
observations for the Missouri, Ohio, and Upper Mississippi River tributaries, and all have simulated discharge magnitudes that are 15–188% larger than observed magnitudes. The Arkansas-Red-White and Lower Mississippi have simulated mean low flows that are seasonally early. The magnitude of simulated Arkansas-Red-White low flows is significantly larger (188%) than USGS observed values, while low flows on the Lower Mississippi are only 43% larger than the observed values (Table 3).

*Seasonality.* We do not expect simulated discharge data to reproduce the actual timing of peak and low flows in an individual year, but we evaluate the ability of CESM1 to skillfully reproduce the average annual seasonality of river discharge. We acknowledge that observed discharge within the Mississippi River basin is influenced by human activities (e.g., reservoirs, levees, irrigation), but note that the seasonal timing of peak flows are minimally impacted at selected gages due to the location of the gages well downstream of high-head dams, or low-head dams which have no significant impact on peak discharges (Remo et
al., 2018). The month of peak flow is, on average, the same pre- and post-modification, or is shifted one month earlier at the Upper Mississippi (Table 2, Figure A1). Additionally, CESM1 output, specifically RTM simulated discharge, has also been used previously to compare directly to gage station data (Dai and Trenberth, 2002). Given the large seasonal offsets between CESM1-simulated and observed discharge, we next turn to hydroclimatic variables to understand why these seasonal offsets in the simulated discharge occur.

**3.2 Hydroclimate variable comparison**

To evaluate the mechanisms that generate the seasonal biases in discharge simulated in CESM1, we examine the major hydrologic variables that contribute to river discharge, including precipitation, total runoff, surface runoff, subsurface drainage,

temperature, evapotranspiration, soil moisture, and snowmelt in both simulations (CESM1) and reanalysis (GPCC and ERA5).
Hydroclimate variables are compared here between CESM1 and reanalysis datasets, rather than point observations such as from

USGS and USACE gages, for data availability and continuity across the domain. Components are discussed below at the
grouped subbasin scale (Eastern, Western, and Entire Mississippi Regions) but results for individual subbasins are included in
appendices (Figure A2 , Figure A3, Table A1).

We find that precipitation and runoff components are the largest contributors to the shift in the seasonality of simulated discharge

(Figure 3) (Table 4). In the CESM1 simulations, seasonal biases in simulations for precipitation, total runoff, surface runoff, and
subsurface runoff, have peak timing differences up to three months. The biases in these four variables culminate in the seasonal
discharge values that are offset from observed values and imply that there are underlying issues in the model that need to be
understood and addressed.

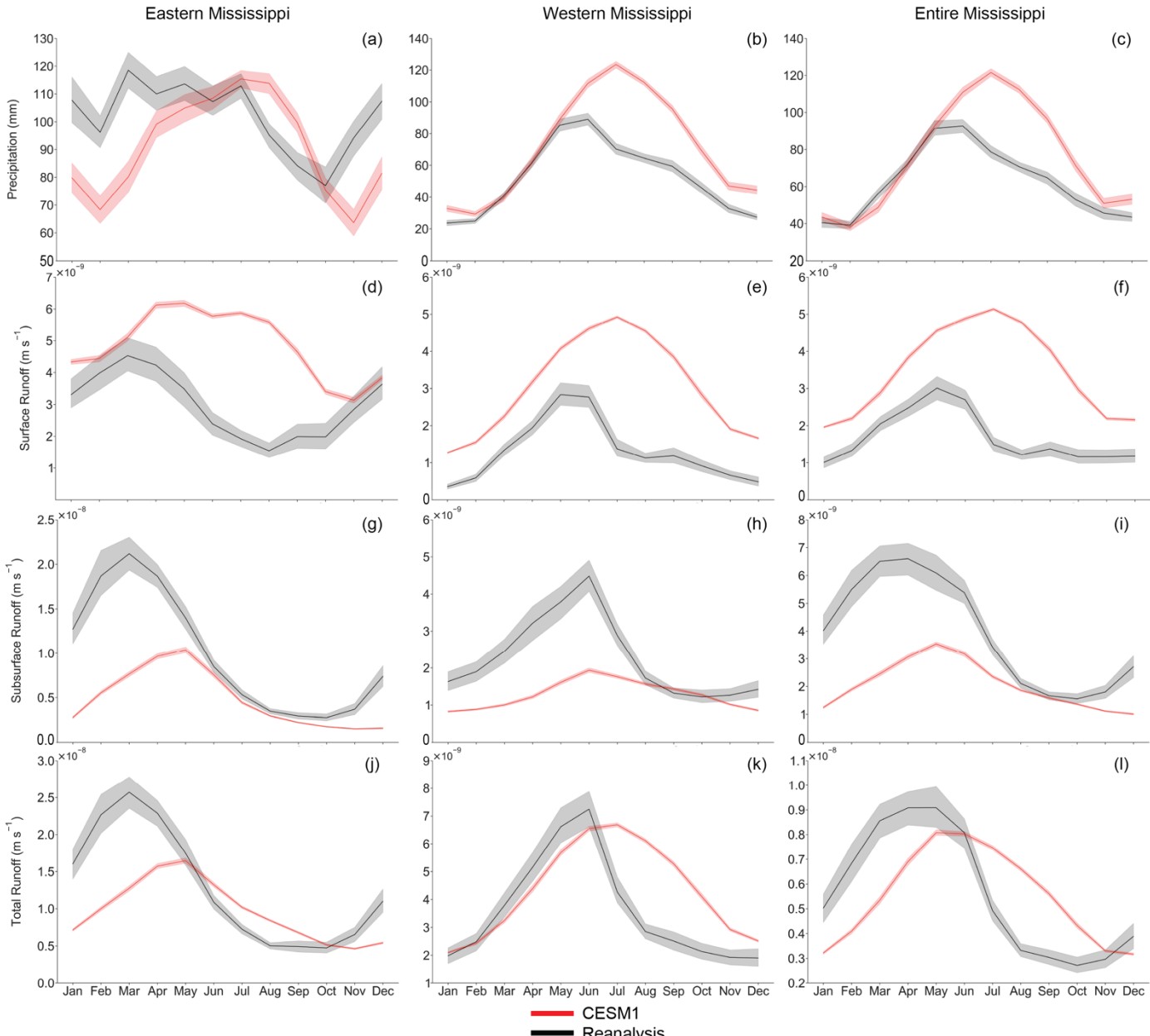

**Figure 3. Monthly mean CESM1 simulated (red) values compared to reanalysis (black) values for precipitation (a-c),
surface runoff (d-f), subsurface runoff (g-i), and runoff (j-l) for the Eastern Mississippi, Western Mississippi, and Entire**

**Mississippi Basin basins. Shading represents a 95% confidence interval from ensemble spread for CESM1 data (50 ensemble members) and interannual variability for both CESM1 and reanalysis data.**

**Table 4. Timing offset and relative difference values for hydroclimate variables between simulated (CESM1) and reanalysis data (ERA5 for surface runoff, subsurface runoff, total runoff; GPCC for precipitation) for maximum and minimum values. Timing offset is in months, where positive values indicate simulated values are delayed relative to reanalysis, and negative values indicate simulated values are early relative to reanalysis. Relative difference values are a percent, and positive values indicate that simulated values are larger, while negative values indicate that simulated values**

**are smaller than reanalysis values.**

| Variable | Basin | Maximum | | Minimum | |
|---|---|---|---|---|---|
| | | Timing offset (months) | Relative Difference (%) | Timing offset (months) | Relative Difference (%) |
| Precipitation | Eastern Mississippi Region | 2 | -6.84 | 1 | -20.03 |
| Precipitation | Entire Mississippi Region | 2 | 29.66 | 1 | 1.26 |
| Precipitation | Western Mississippi Region | 1 | 40.36 | 1 | 30.34 |
| Surface Runoff | Eastern Mississippi Region | 2 | 35.98 | 3 | 124.43 |
| Surface Runoff | Entire Mississippi Region | 2 | 71.65 | 0 | 100.48 |
| Surface Runoff | Western Mississippi Region | 2 | 76.92 | 0 | 267.12 |
| Subsurface Runoff | Eastern Mississippi Region | 2 | -51.31 | 1 | -44.55 |
| Subsurface Runoff | Entire Mississippi Region | 1 | -47.39 | 2 | -31.65 |
| Subsurface Runoff | Western Mississippi Region | 0 | -57.58 | 3 | -31.26 |
| Total Runoff | Eastern Mississippi Region | 2 | -36.11 | 1 | 8.03 |
| Total Runoff | Entire Mississippi Region | 1 | -10.85 | 2 | 25.09 |
| Total Runoff | Western Mississippi Region | 1 | -7.96 | 1 | 11.83 |

*Precipitation*: CESM1 simulated precipitation is seasonally delayed in the Western, Eastern, and across the Entire Mississippi

basin for both peak (1-2 months) and minimum (1 month) values when compared to reanalysis data. Simulated precipitation has a magnitude larger than reanalysis in the Western and Entire Mississippi basins for both the peak (29.66 – 40.36%) and minimum (1.26 - 30.34%) values, but the magnitude is smaller in the Eastern Mississippi basin for both the peak (-6.84%)  and minimum values (-20.03%) (Figure 3a-c) (Table 4). The delayed timing of simulated precipitation causes the peak to occur in July across all portions of the basin, up to two months after the peak in reanalysis data, and during summer months when peak

rainfall is less likely to occur in this climate.

*Surface runoff*: Similar to precipitation, CESM1 simulated peak surface runoff is delayed relative to ERA5 reanalysis across all basins of the Mississippi River basin (2 months) (Figure 3d-f) (Table 4). Minimum surface runoff is only delayed in the Eastern Mississippi basin relative to ERA5 (3 months), but timing is aligned in the Western Mississippi and across the Entire basin. In all

basins examined here, the magnitudes of simulated peak and minimum runoff are larger ( 35 – 267%) than those of the peak and minimum runoff values in reanalysis data. Patterns in the time series shape for surface runoff reflect the seasonal precipitation patterns in CESM1, suggesting precipitation plays a role in the delayed timing of runoff: in the Eastern region of the basin,

simulated surface runoff peaks two months after the peak in reanalysis data and instead of immediately declining, following the shape of the runoff reanalysis time series shape, CESM1 simulated runoff remains near its peak from June through August before declining in the fall. CESM1 surface runoff in the Western basin and Entire Mississippi basin similarly resembles the shape of the CESM1 precipitation time series, and declines from peak values more gradually than the reanalysis time series. All three basins mimic the shape of the CESM1 precipitation time series, rather than the reanalysis time series of surface runoff (Figure 3d-f).

*Subsurface runoff*: Subsurface runoff is seasonally delayed in both the Eastern Mississippi and across the entire Mississippi basin (1-3 months), but the peak for maximum subsurface runoff is aligned for the Western Mississippi basin when CESM1 simulations are compared to ERA5 (Figure 3g-i) (Table 4). Simulated seasonal peak and minimum magnitudes of CESM1 data are smaller (-31.26 – -57.58%) for all basins than peak and minimum magnitudes of subsurface drainage in reanalysis data. While the timing of peak values is aligned in the Western Mississippi basin, CESM1 subsurface runoff values decline more gradually than reanalysis values. The shape of CESM1 and reanalysis time series in the Eastern Mississippi and across the Entire Mississippi Basin are more similar as subsurface runoff values decline from their peak.

*Total runoff*: CESM1 simulated total runoff is delayed across all basins relative to ERA5 reanalysis for both peak and minimum values (1-2 months) (Figure 3j-l) (Table 4). Magnitudes of CESM1 peak values are smaller than those of ERA5 total runoff (-7.96 – -36.11%), while minimum values are larger than ERA5 values (8.08 – 25.09%). In the Western Mississippi basin (Figure 3k), the shape of the time series more closely resembles the time series of surface runoff and precipitation from the Western Mississippi basin than the total runoff in the reanalysis time series. In the Eastern Mississippi (Figure 3j), the shape of the total runoff time series reflects the shape of the subsurface time series. At the scale of the Entire Mississippi (Figure 3l), total runoff resembles subsurface runoff from January through its peak in May, but is more similar to surface runoff as it declines through December.

Precipitation biases previously documented in other regions and the Mississippi River Basin are primarily due to regional scale processes including deep convection parameterization or low-level moisture divergence and convergence (Benedict et al., 2017; Moseley et al., 2016; Sakaguchi et al., 2018; Wang and Zhang, 2016), the impacts of modeled climate teleconnections on simulated precipitation, due to the climate forcings used, which include precipitation (Li et al., 2015), model resolution, or as documented in experimental setups (Li et al., 2013, 2015). However, a precipitation bias has not been previously documented over the Mississippi Basin in CESM1, and is the most significant driver of the shift in timing of the simulated discharge. This bias propagates through to surface runoff, particularly in the Eastern Mississippi basin where rainfall dominates the hydrologic cycle.

Additionally, subsurface runoff is impacted by the routing mechanisms in the River Transport Model (RTM) of CESM1. While the model has been shown to accurately simulate runoff and discharge for small watersheds (<66,000 $km^2$), there are biases due to the routing, which becomes more severe the larger the watershed (Li et al., 2013). This is relevant for the Mississippi River basin as its drainage area is ~3.2 million $km^2$. Prior work has shown that RTM overestimates the time lag from surface runoff generation to when it appears as discharge, especially for larger watersheds. The RTM also assumes homogeneity between grid cells and a constant channel velocity (Li et al., 2013), both of which hinder the models ability to fully capture seasonal and spatial variability.

*Other hydrologic variables*: Seasonal biases are less pronounced in temperature, evapotranspiration, soil moisture, and snowmelt
(Figure 4) (Table 5), with peak timing differences of zero to two months.

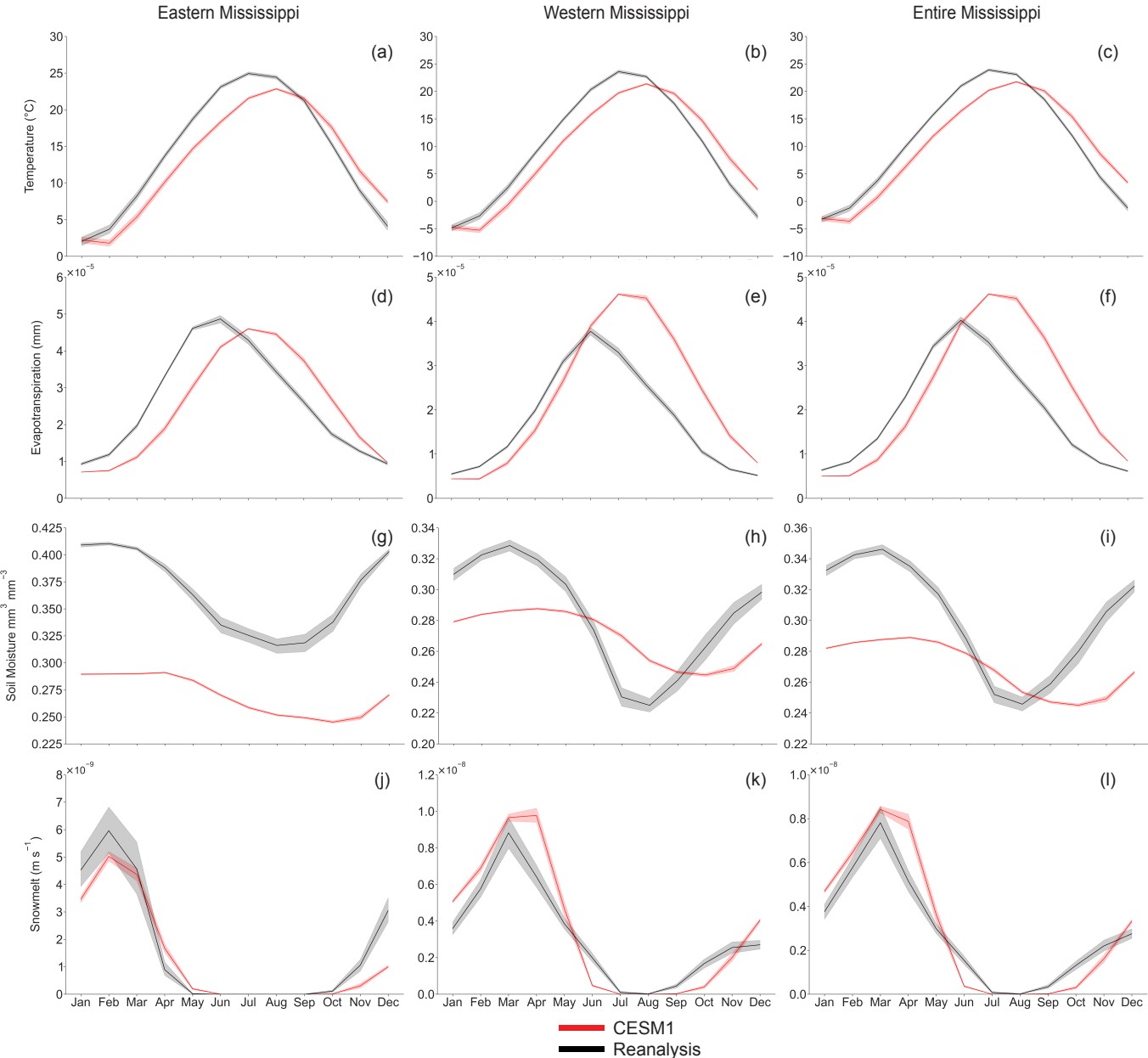

**Figure 4. Monthly mean CESM1 simulated (red) values compared to reanalysis (black) values for temperature (a-c), evapotranspiration (d-f), soil moisture (g-i), and snow melt (j-l) for the Eastern Mississippi, Western Mississippi, and Entire Mississippi Basin basins. Shading represents a 95% confidence interval from ensemble spread for CESM1 data**
**(50 ensemble members) and interannual variability for both CESM1 and reanalysis data.**

**Table 5. Timing offset and relative difference values for hydroclimate variables between simulated (CESM1) and reanalysis data (ERA5 for temperature, soil moisture, and snowmelt; Livneh for evapotranspiration) for maximum and minimum values. Timing offset is in months, where positive values indicate simulated values are delayed relative to**
**reanalysis, and negative values indicate simulated values are early relative reanalysis. Relative difference values are a**

**percent, and positive values indicate that simulated values are larger, while negative values indicate that simulated values are smaller than reanalysis values.**

| Variable | Basin | Timing offset (months) | Relative Difference (%) | Timing offset (months) | Relative Difference (%) |
|---|---|---|---|---|---|
| Temperature | Eastern Mississippi Region | 1 | -7.93 | 1 | 8.05 |
| Temperature | Entire Mississippi Region | 1 | -7.93 | 1 | 3.20 |
| Temperature | Western Mississippi Region | 1 | -8.16 | 1 | 4.52 |
| Evapotranspiration | Eastern Mississippi Region | 1 | -6.12 | 0 | -22.22 |
| Evapotranspiration | Entire Mississippi Region | 1 | 17.50 | 1 | -16.67 |
| Evapotranspiration | Western Mississippi Region | 1 | 23.68 | 1 | -20.00 |
| Soil Moisture | Eastern Mississippi Region | 2 | -29.13 | 2 | -23.31 |
| Soil Moisture | Entire Mississippi Region | 1 | -17.12 | 2 | -0.93 |
| Soil Moisture | Western Mississippi Region | 1 | -13.22 | 2 | 8.21 |
| Snowmelt | Eastern Mississippi Region | 0 | -20.89 | 2 | 0 |
| Snowmelt | Entire Mississippi Region | 0 | 2.58 | 0 | 0 |
| Snowmelt | Western Mississippi Region | 1 | 5.86 | 0 | 0 |

CESM1 maximum and minimum temperature values are one month late relative to reanalysis data. CESM1 maximum values are all smaller than reanalysis values (-7.93 – -8.16%), while minimum values are all larger than reanalysis values (3.20 – 8.05%) (Figure 4a-c) (Table 5).

CESM1 evapotranspiration is one month late relative to reanalysis in the Eastern and Western basins, and one month early for the entire Basin when peak values are examined. Minimum values are aligned in the Eastern Mississippi basin, and one month
late in the Western and Entire Mississippi basins. The maximum values of evapotranspiration in the Western Mississippi basin and Entire Mississippi basin are larger than reanalysis values (17.50 – 23.68%), all other minimum and maximum values are smaller than reanalysis values (-6.12 – -22.22%) (Figure 4d-f) (Table 5).

Soil moisture in CESM1 simulations is one to two months late for all basins for both minimum and maximum values relative to
reanalysis data. All CESM1 values are smaller than reanalysis data (-0.93 – -29.13%) except for the minimum value in the Western Mississippi basin (8.12%) (Figure 4g-i) (Table 5).

Snowmelt has no difference in timing of the maximum values for the Eastern Mississippi or Entire Mississippi Basins, or the minimum values of the Western Mississippi or Entire Mississippi basin. The CESM1 Western Mississippi peak and Eastern
Mississippi minimum values are late relative to reanalysis (1-2 months) (Figure 4j-l) (Table 5). CESM1 snowmelt maximum values in the Entire Mississippi basin and Western Mississippi basin are larger than reanalysis (2.58 – 5.86%), and smaller in the Eastern Mississippi basin (-20.89%). Minimum values for all basins approach zero. (Table 3).

Temperature, evapotranspiration, soil moisture, and snowmelt are less impacted by precipitation and are better represented by
CESM1. Evapotranspiration can be impacted by rainfall, however it is also governed by solar radiation, wet leaf fraction, canopy evaporation, and vegetation transpiration (Cui et al., 2022). The snow model is noted as being an area of new improvement in CESM1, with updates to modeled snow cover and related parameterizations (Lawrence et al., 2011). Of note, snow melt has not

been independently validated, though other variables related to snow processes have and generally perform well (Cammalleri et al., 2022; Kouki et al., 2023; Tarek et al., 2020). Lastly, soil moisture has been evaluated in other contexts and shown to perform
well in CESM1 across CONUS at different soil depths, so the skill here is consistent with previous findings (Yuan and Quiring, 2017). Overall we expect temperature, evapotranspiration, soil moisture, and snowmelt to be skillful based on the model setup and governing factors, and the analysis supports this.

**3.3 Relative Difference**

Relative difference values vary widely between regions and across variables. CESM1 values are larger than all reanalysis values
of surface runoff (35.98 – 267.12%) and smaller than all ERA5 subsurface runoff values (-31.26 – -57.58%). Total runoff has CESM1 simulated maximum values that are smaller than ERA5 values (-7.96 – -36.11%), and CESM1 simulated minimum values that are larger than ERA5 values (8.03 – 25.09%). Precipitation values are larger in CESM1 simulations for the Entire Mississippi and Western Mississippi (1.2 – 40.36%) and smaller in the Eastern Mississippi (-6.84 – -20.03%) (Table 4).  Relative differences tended to be smaller between CESM1 simulated values and ERA5 values for hydrologic variables with less pronounced seasonal
biases (-29.13 – 23.68%) (Table 5). Differences in magnitudes can be bias corrected (Teutschbein and Seibert, 2012), so offsets in timing rather than magnitudes are further investigated.

**3.4 Lag Correlation**

Lag correlation indicates the timing offset at which two time series are best correlated, and here supports our comparison of the monthly mean time series (Figure 4a-h, Table A2). Temperature, evapotranspiration, soil moisture, and snow melt exhibit peak
correlations at no lag (0 months) or slightly delayed (-1 month) when CESM1 is compared to reanalysis data, supporting our assessment that these variables are simulated relatively skillfully in CESM1 (Figure 5e-h). Soil moisture is highly correlated when the time series are not lagged. The Eastern Mississippi Basin soil moisture has a maximum correlation with no lag (0 months). However the maximum correlations for soil moisture are at negative seven months for the Western and Entire Mississippi basins, but the second highest correlation values for these two basins are at a lag of zero months. Evapotranspiration
has a peak correlation for both grouped subbasins at negative one month, but a maximum correlation for the entire basin when there is no lag. Temperature has a maximum correlation for all three basin groupings at a lag of negative one month. Snowmelt has a maximum correlation for all three basin groupings when there is no lag between the CESM1 and reanalysis time series.

In contrast, precipitation has lag correlations that support simulated peak values being offset from reanalysis peak values,
particularly in the Eastern Mississippi Basin (Figure 5a). The peak correlation of precipitation, particularly in the Eastern Mississippi Basin, supports the timing of precipitation being a factor in the delayed runoff and discharge. Correlation values of precipitation are at a maximum for the Eastern Mississippi basin at a lag of negative five months, and for the Western and entire Mississippi basin at a lag of negative one month.

Peak correlations of all three runoff variables (subsurface runoff, surface runoff, total runoff; Figure 5b-d) both support their contributions to, and align with the previous findings that the RTM model has biases due to the runoff routing (Li et al., 2013, 2015). Surface runoff has a peak correlation at negative seven months for the Eastern Mississippi basin, and at negative one month for the Western and Entire Mississippi basin. Subsurface runoff has a peak correlation of negative one month for all basins, and total runoff has a peak correlation at negative one month for the Western Mississippi Basin, and negative two months

for the Eastern and Entire Mississippi basin. However, both have bimodal lag correlation peaks for the Eastern basin, where a second peak correlation is at a lag of negative seven months.

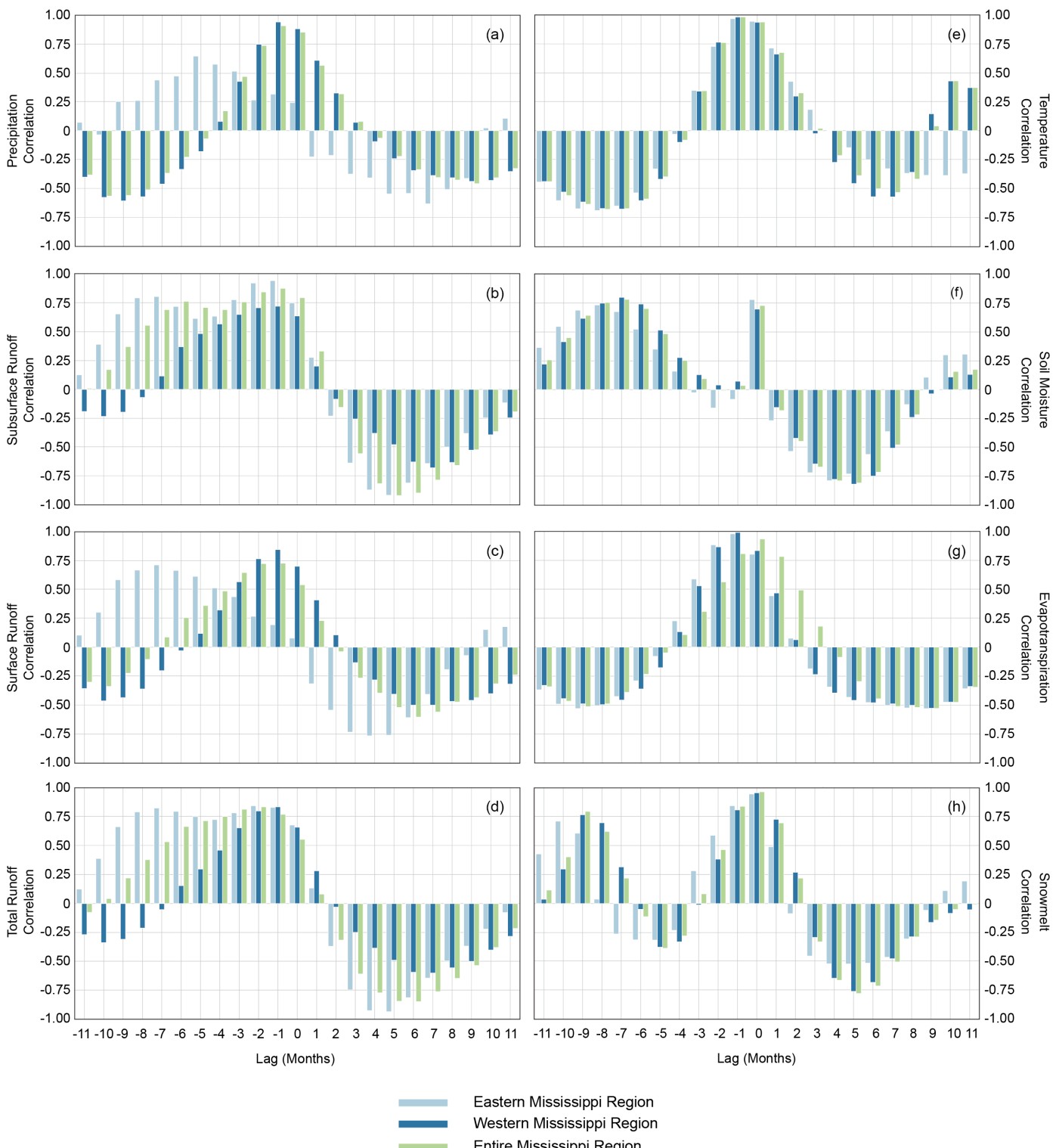

**Figure 5. Monthly lag correlation values for each hydrologic variable [a) precipitation, b) subsurface runoff, c) surface runoff, d) runoff, e) temperature, f) soil moisture, g) ET, h) snowmelt], for each region: Eastern Mississippi basin (light**
**blue), Western Mississippi Basin (dark blue), Entire Mississippi basin (green).**

### 3.5 Spectral Angle

Spectral angle (Figure 6) is used to compare the shape of the time series without comparing the magnitude or timing offsets. It treats the data being compared as dimensionless unit vectors to assess if they have the same direction in space, which indicates similarity in shape regardless of similarity in magnitude. A value closer to zero indicates better agreement between the shape of two time series. A value of zero would mean that one vector, or time series in this case, was identical to the other in its shape (Jackson et al., 2019).

Temperature, evapotranspiration, and soil moisture have the lowest values, or best agreement. Soil moisture has the lowest values (0.098 – 0.115), and temperature (0.266 – 0.391) and evapotranspiration (0.256 – 0.411) values fall within similar ranges across the Eastern Mississippi, Western Mississippi, and Entire Mississippi Basin basins (Table A3).

Conversely, subsurface runoff, surface runoff, total runoff, and precipitation have the highest values across all basins, indicating worse agreement. Subsurface drainage (0.543 – 0.640), surface drainage (0.576 – 0.681), and runoff (0.545 – 0.604) have spectral angle values that fall within similar ranges across the basin groupings. The range of precipitation values (0.378 – 0.388) falls below those of runoff related variables, while snowmelt has the widest range (0.474 – 0.733) (Table A3).

For all variables other than temperature and soil moisture, values are lower for the entire basin than at the grouped basin scale. For both temperature and soil moisture, the Eastern Mississippi has a lower value. Conversely, snowmelt has a significantly higher value in the Eastern Mississippi basin.

These spectral angle values help demonstrate that while the seasonality is severely shifted for several hydroclimate variables, the shape of the annual time series for other CESM1 variables is similar between simulated and reanalysis datasets. For temperature, evapotranspiration, and soil moisture, the spectral angle values, along with lag correlation and relative bias, are small, suggesting these variables represent average annual seasonality relatively well. For precipitation, subsurface drainage, surface drainage and total drainage, the higher values of spectral angle show that in addition to the seasonal timing offset, the shape of the time series is not well represented. This supports the precipitation biases and limitations of the RTM in simulating runoff related variables over large basins.

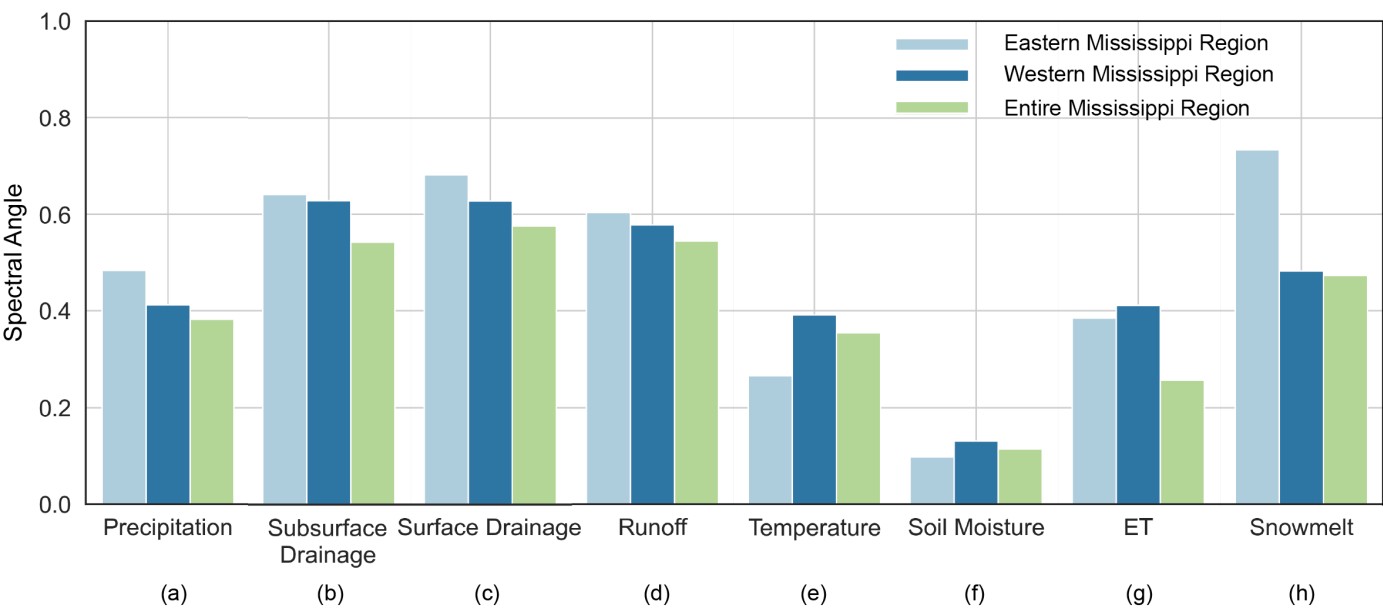

Figure 6. Spectral angle (SA) values for each hydrologic variable [a) precipitation, b) subsurface runoff, c) surface runoff, d) runoff, e) temperature, f) soil moisture, g) ET, h) snowmelt], for each region: Eastern Mississippi Basin (light blue), Western Mississippi Basin (dark blue), Entire Mississippi basin (green).

### 3.5 CMIP6 model runoff comparison

Finally, we compare how other CMIP6 models perform in simulating runoff over the Mississippi River basin relative to reanalysis (Figure 7). The comparison of runoff is necessary as only CESM1 and CESM2 explicitly simulate river discharge, but we expect biases in the magnitude and seasonality of runoff to closely mirror those of river discharge. In general, the timing of runoff in some CMIP6 models shows better agreement with ERA5 reanalysis compared to CESM1; the timing of modeled runoff is also more accurately captured by CMIP6 models in the Eastern Mississippi Basin than by CESM1. Notably, modeled runoff in CESM2 has improved timing of maximum and minimum flows relative to ERA5. Additional figures with runoff comparison of CMIP6 model, CESM1, and ERA5 by individual subbasin are included in Figure A4.

Seasonal maxima of runoff are most closely aligned between models in the Eastern basin of the Mississippi basin, and have more spread but still overall agreement between most models in the Western basin of the Mississippi Basin and across the entire Mississippi Basin. In the Eastern basin, CESM2, BCC CSM2 MR, CanESM5, MIROC6, and MPI ESM1 2 LR all peak in March, aligning with ERA5 reanalysis, as opposed to CESM1, which peaks in May. In the Western Mississippi basin, CESM2 peak runoff occurs in March, as is the case for BCC CSM2 MR, CanESM5, and MIROC6; ERA5 peaks in May, and CESM1 peaks in July. At the Entire Mississippi River Basin scale, CESM2, BCC CSM2 MR, CanESM5, MIROC6, and MPI ESM1 2 LR have peaks aligned in March, and ERA peaks in April and May.

Runoff minimums have more spread in timing between CMIP6 models. In the Western region of the basin, ERA5 has a minimum runoff in November, whereas other models have minimums in August, September, and December. In the Eastern basin, ERA5 reaches a minimum in October, as does CanESM5, where BCC CSM2 MR has a minimum in August, CESM2, MPI ESM1 2 LR and MRI ESM2 0 reach minimums in October, and CESM1 reaches a minimum in November. Across the Entire Mississippi Basin, ERA5 reaches a minimum in October, with all other models again ranging between August and December.

Overall, our findings show that CMIP6 models exhibit improvements in the seasonal timing of runoff compared to CESM1. A major benefit of CESM1, however, is that it is one of the few CMIP5 models that has a routing model and multiple available modeling projects, including the Large Ensemble (CESM-LE) (Kay et al., 2015) and the Last Millennium Ensemble (CESM-475 LME) (Otto-Bliesner et al., 2016). Of note, our findings highlight the improvements of CESM2 over CESM1, which is due to updates to the routing model, namely from the RTM to the Model for Scale Adaptive River Transport (MOSART) model (Li et al., 2013, 2015). The MOSART model uses the kinematic wave equation to simulate streamflow, improving hydrograph timing and values over RTM. Additionally, MOSART incorporates spatial heterogeneity across grid cells, whereas RTM uses spatial homogeneity with the assumptions of spatially uniform constant velocity, allowing MOSART to perform better across spatial 480 scales. Overall, MOSART has been shown to better capture the time lag between runoff generation and streamflow, a critical issue also demonstrated in the Mississippi River Basin here with CESM1(Li et al., 2015).

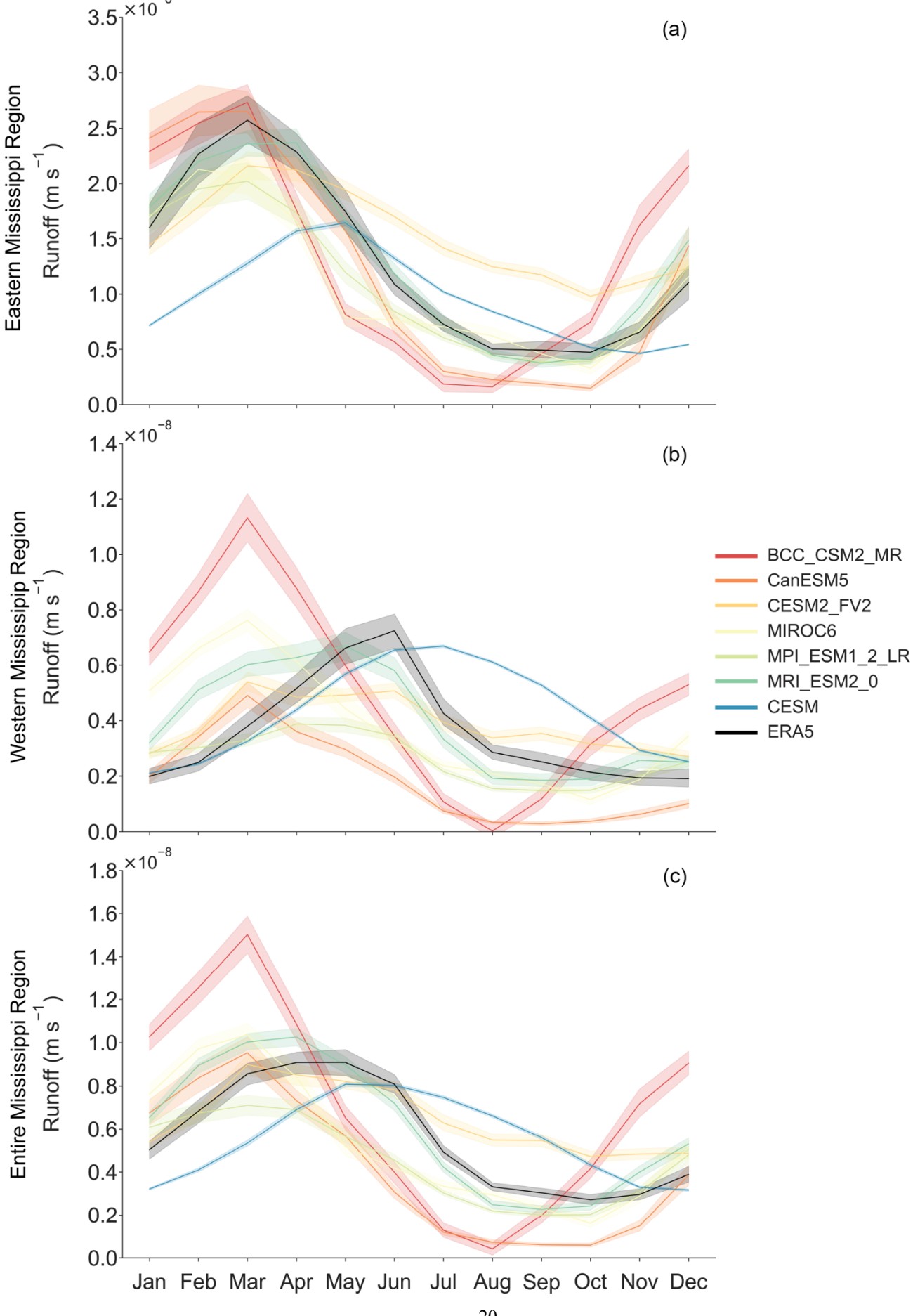

**Figure 7. Monthly mean simulated runoff (m/s) from selected CMIP6 models (BCC CSM2 MR [red], CanESM5 [dark orange], CEMS2 [light orange], MIROC6 [yellow], MPI ESM1 2 LR [light green], MRI ESM2 0 [dark green]), CESM1 [blue], and ERA5 reanalysis [black] for the Western Mississippi basin (a), Eastern Mississippi basin (b), and Entire Mississippi Basin (c). Shading represents a 95% confidence interval from interannual variability.**

**4 Conclusions**

In this study, we investigated the skill of CESM1 in simulating hydrologic processes over the Mississippi River basin. This model (CESM1) is unique among CMIP models because it simulates river discharge, and has been used for understanding the hydrologic changes in the Mississippi basin in the past and future (Munoz and Dee, 2017; Wiman et al., 2021). Our analysis shows that CESM1-simulated river discharge exhibits large biases in both its magnitude and seasonality relative to stream gage measurements. The causes of this seasonal bias were diagnosed by comparing simulations to reanalysis products (GPCC, ERA5); we showed that the seasonal bias arises primarily from the delayed timing of precipitation and runoff related processes in CESM1. Simulated precipitation, surface runoff, subsurface runoff, and total runoff are all delayed relative to reanalysis data by up to three months. An examination of runoff over the Mississippi River basin in several CMIP6 models including BCC CSM2 MR, CanESM5, CESM2, MIROC6, MPI ESM1 2 LR, and MRI ESM2 0 reveals simulated runoff seasonality is more aligned with reanalysis than that in CESM1. Of note, the seasonality of CESM2 simulated runoff exhibits significant improvement relative to CESM1. This is due to a major update in the river routing model from the River Transport Model (RTM) in CESM1 to MOSART in CESM2. Our analysis implies that CESM1 discharge, runoff, and precipitation should be used with caution over the Mississippi River basin, but that temperature, evapotranspiration, soil moisture, and snowmelt perform relatively well. We also show significant improvement in runoff simulations from CESM1 to CESM2 over the Mississippi River basin, implying that discharge simulations from CESM2 provide a more accurate projection of future hydroclimate conditions in the basin, and should thus be prioritized in future analyses.

The improvements in surface runoff noted here from a CMIP5 model (CESM1) to a suite of CMIP6 models represents a broader progress in the representation of surface water hydrology in earth system models (Pokhrel et al., 2016). Robust simulations of hydrologic processes — especially river discharge — in earth system models is of critical importance for effective management of water resources. Yet, relatively few CMIP6 models simulate river discharge directly, resulting in the use of other variables related to discharge (e.g., precipitation, runoff), or in the development of hydrologic models to explicitly simulate river flows offline. When hydrologic models are developed separately, understanding the seasonal biases in all variables that may be used to simulate river flows is critical to assessing if output is adequate for applications such as practical risk assessment. Ideally, river discharge would be skillfully modeled as part of all CMIP models to provide standardized output that could be used by water resource managers and other stakeholders to evaluate projected changes in water resources. As these models continue to add complexity in their representation of surface water hydrology, we encourage further inclusion of human interventions in hydrologic processes, including large reservoirs, channelization, and agricultural and municipal water use. Comprehensive and skillful simulations of streamflow for large and economically important river systems, including the Mississippi River basin, are of critical importance. Our study represents a first step towards validation of available earth system model simulations of Mississippi River basin hydrology, and provides a foundation from which robust analyses of past and projected changes in river discharge can emerge.

**Code Availability**

All code necessary for reproducing the results is provided at https://doi.org/10.5281/zenodo.11211748 (O'Donnell, 2024).

**Data Availability**

USGS discharge data is available from https://waterdata.usgs.gov/nwis. US Army Corps of Engineers discharge data for the
525 Mississippi River at Tarbert Landing can be accessed via
https://rivergages.mvr.usace.army.mil/WaterControl/stationinfo2.cfm?sid=01100Q&dt=%20S. CESM1 data can be retrieved
from https://www.cesm.ucar.edu/community-projects/lme/data-sets. ERA5 reanalysis can be accessed via
https://cds.climate.copernicus.eu/cdsapp#!/dataset/reanalysis-era5-single-levels-monthly-means?tab=form, GPCC data via
https://iridl.ldeo.columbia.edu/SOURCES/.WCRP/.GCOS/.GPCC/.FDP/.version2018/.2p5/.prcp/datafiles.html, and the Livneh
Hydrometeorological dataset via https://psl.noaa.gov/data/gridded/data.livneh.html. CMIP6 data is available from
https://cds.climate.copernicus.eu/cdsapp#!/dataset/projections-cmip6?tab=form.

**Author Contributions**

S.M. and S.D. initiated the project. M.O. performed analyses and analyzed data. M.O. wrote the manuscript with contributions to
writing, review, and editing from K.M., J.D.G, S.D., and S.M.

**Declaration of competing interest**

The authors declare that they have no conflict of interest.

**Acknowledgements**

This study was funded by the U.S. National Science Foundation Division of Atmospheric and Geospace Sciences (award number
2147782). Additional support to M.O was provided by the National Science Foundation Graduate Research Fellowship (DGE
1650115).

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

**Appendices**

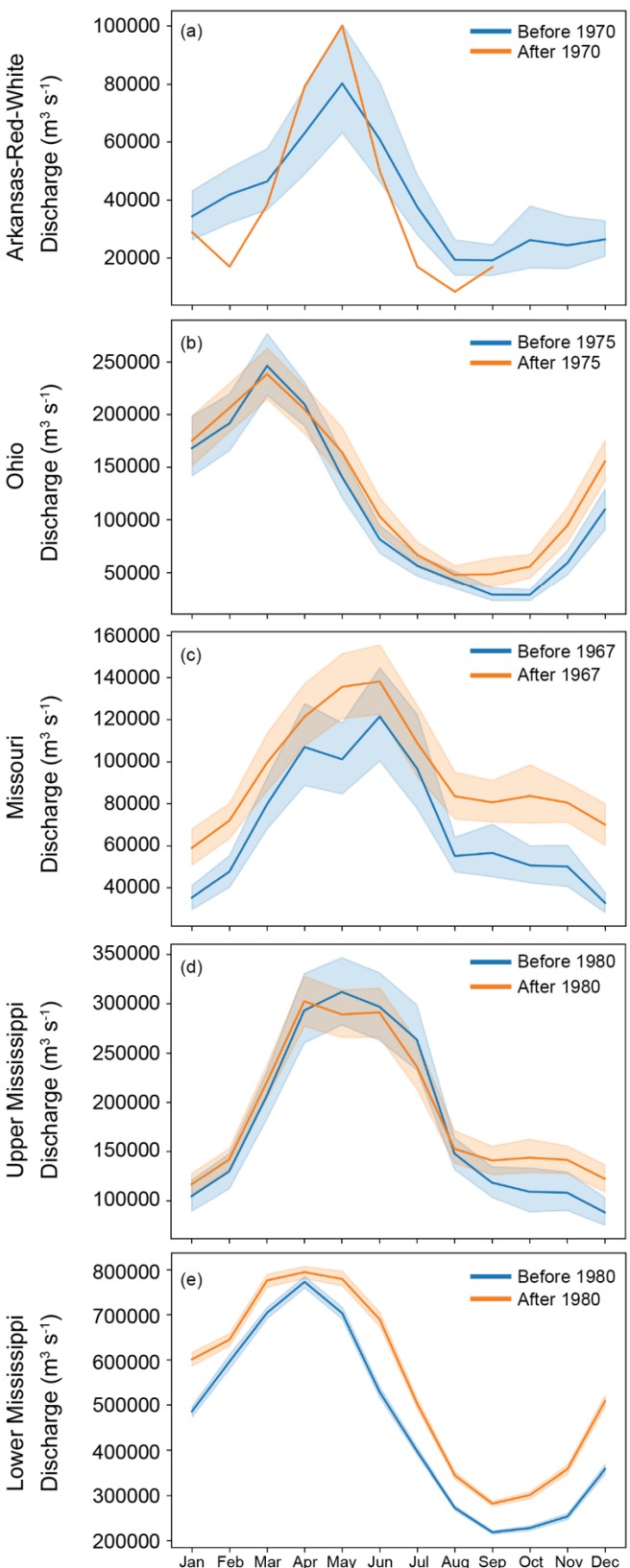

**Figure A1: Mean monthly discharge values before and after periods of significant dam construction and river**
**engineering for the tributaries a) Arkansas-Red-White, b) Ohio, c) Missouri, d) Upper Mississippi, e) Lower Mississippi.**
**Shading represents a 95% confidence interval from interannual variability.**

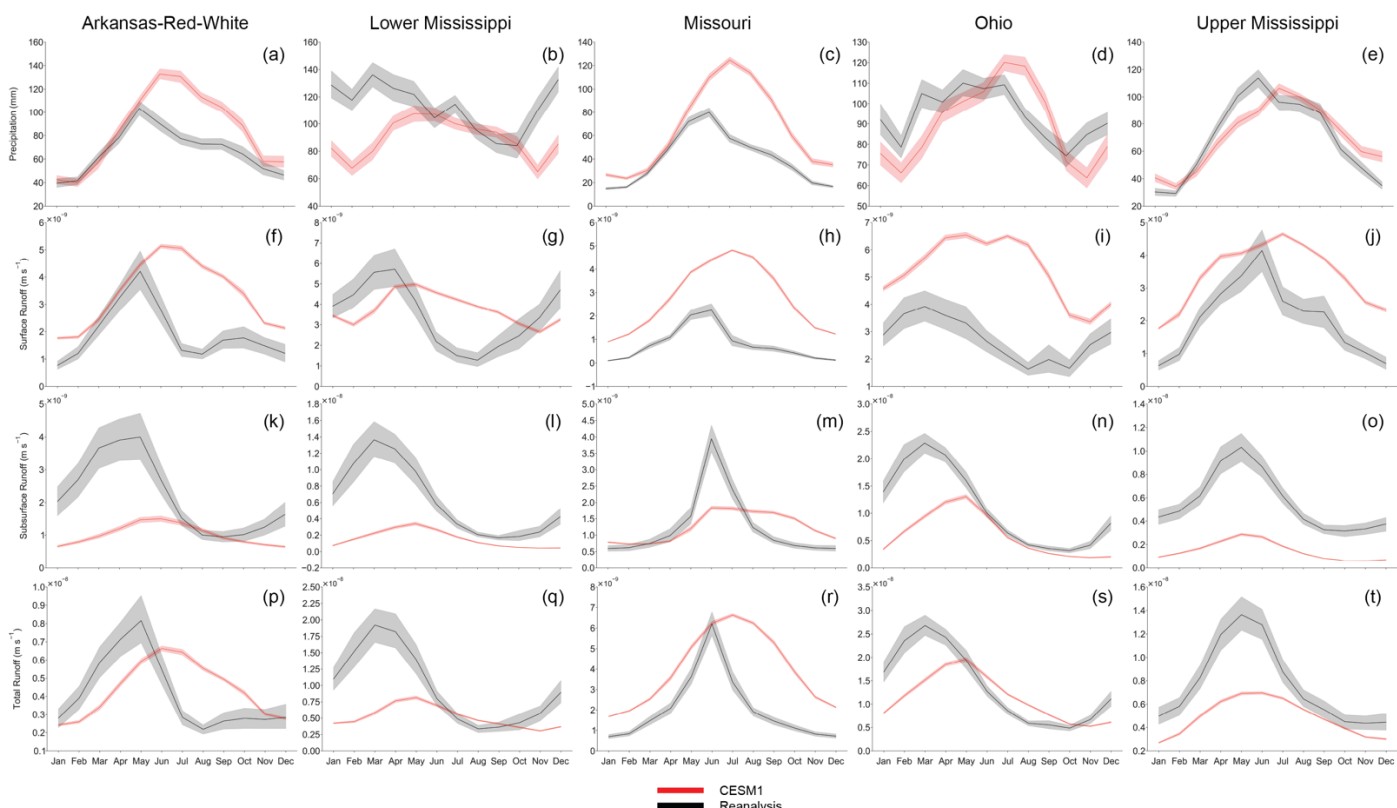

**Figure A2: Monthly mean CESM1 simulated (red) values compared to reanalysis (black) values for precipitation (a-e), surface runoff (f-j), subsurface runoff (k-o), and runoff (p-t) for the major subbasins: Arkansas-Red-White, Lower Mississippi, Missouri, Ohio, Upper Mississippi. Shading represents a 95% confidence interval from ensemble spread for CESM1 data (50 ensemble members) and interannual variability for both CESM1 and reanalysis data.**

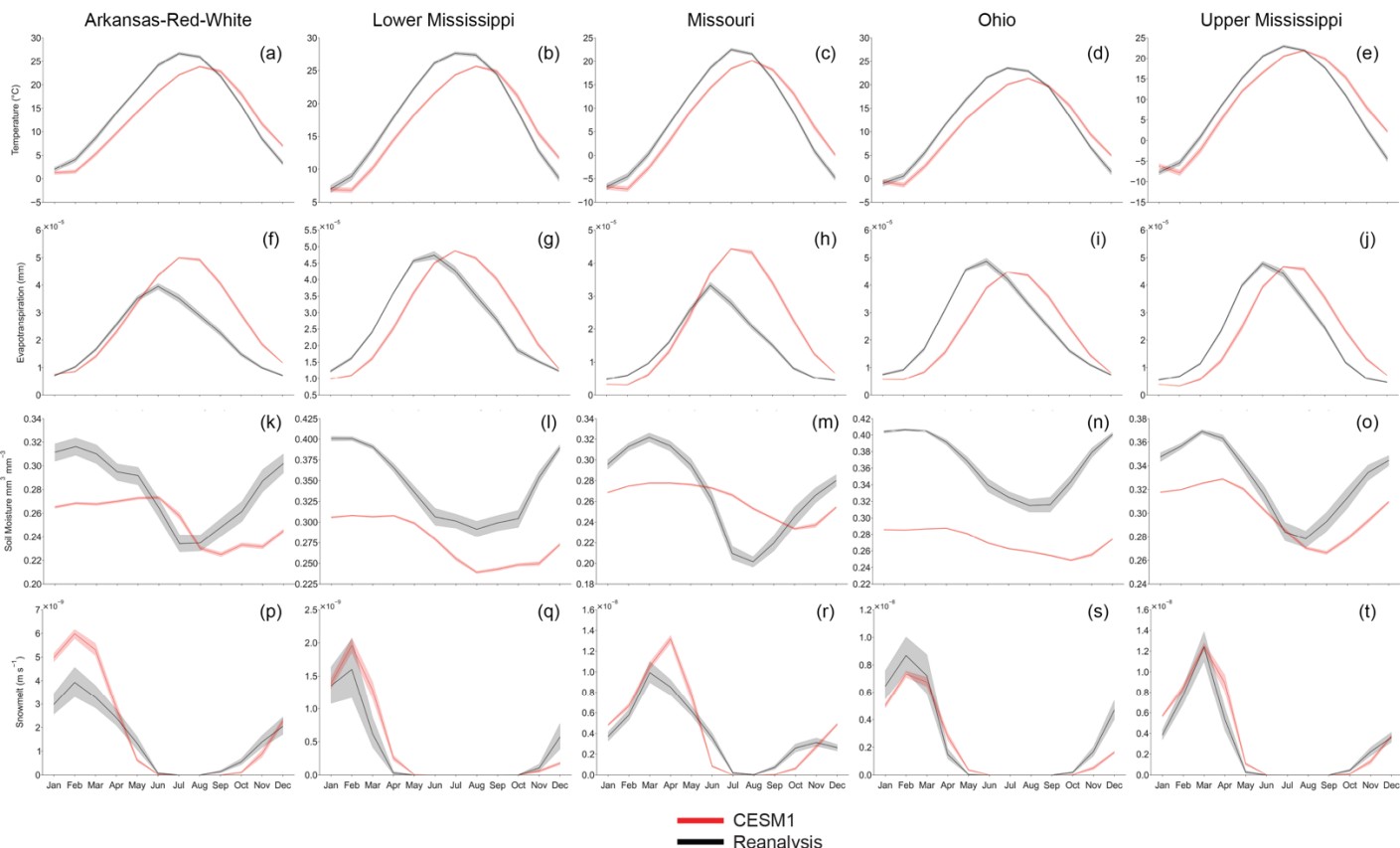

**Figure A3: Monthly mean CESM1 simulated (red) values compared to reanalysis (black) values for temperature (a-e), evapotranspiration (f-j), soil moisture (k-o), and snowmelt (p-t) for the major subbasins: Arkansas-Red-White, Lower Mississippi, Missouri, Ohio, Upper Mississippi. Shading represents a 95% confidence interval from ensemble spread for CESM1 data (50 ensemble members) and interannual variability for both CESM1 and reanalysis data.**


**Table A1: Timing offset and relative difference values for hydroclimate variables between simulated (CESM1) and reanalysis data (ERA5 for temperature, surface runoff, subsurface runoff, total runoff, soil moisture, and snowmelt; GPCC for precipitation; Livneh for evapotranspiration) for maximum and minimum values for all subbasins (Arkansas-Red-White, Lower Mississippi, Missouri, Ohio, and Upper Mississippi). Timing offset is in months, where positive values indicate simulated values are delayed relative to reanalysis, and negative values indicate simulated values are early relative reanalysis. Relative difference values are a percent, and positive values indicate that simulated values are larger, while negative values indicate that simulated values are smaller than reanalysis values.**

| Variable | Basin | Maximum | | Minimum | |
|---|---|---|---|---|---|
| | | Timing offset (months) | Relative Difference (%) | Timing offset (months) | Relative Difference (%) |
| Temperature | Arkansas-Red-White | 1 | -9.76 | 0 | -26.64 |
| Temperature | Lower Mississippi | 1 | -6.67 | 1 | 1.41 |
| Temperature | Missouri | 1 | -8.85 | 1 | 3.68 |
| Temperature | Ohio | 1 | -9.08 | 1 | -4.68 |
| Temperature | Upper Mississippi | 1 | -3.72 | 1 | 2.72 |
| Precipitation | Arkansas-Red-White | 1 | 26.17 | 1 | 5.83 |
| Precipitation | Lower Mississippi | 1 | -19.55 | 1 | -24.86 |
| Precipitation | Missouri | 1 | 60.80 | 1 | 68.17 |
| Precipitation | Ohio | 2 | 1.92 | 1 | -16.44 |
| Precipitation | Upper Mississippi | 1 | -5.96 | 0 | 23.51 |
| Surface Runoff | Arkansas-Red-White | 1 | 19.27 | 0 | 126.69 |
| Surface Runoff | Lower Mississippi | 1 | -13.71 | 3 | 169.44 |
| Surface Runoff | Missouri | 1 | 116.29 | 0 | 847.45 |
| Surface Runoff | Ohio | 2 | 67.40 | 1 | 127.37 |
| Surface Runoff | Upper Mississippi | 1 | 25.22 | 1 | 190.88 |
| Subsurface Runoff | Arkansas-Red-White | 2 | -64.55 | 3 | -35.20 |
| Subsurface Runoff | Lower Mississippi | 2 | -78.60 | 1 | -74.53 |
| Subsurface Runoff | Missouri | 0 | -54.39 | 2 | 20.48 |
| Subsurface Runoff | Ohio | 2 | -41.73 | 1 | -40.05 |
| Subsurface Runoff | Upper Mississippi | 0 | -71.61 | 1 | -80.34 |
| Total Runoff | Arkansas-Red-White | 1 | -21.16 | -7 | 8.38 |
| Total Runoff | Lower Mississippi | 2 | -59.97 | 3 | 3.34 |
| Total Runoff | Missouri | 1 | 7.00 | 0 | 139.25 |
| Total Runoff | Ohio | 2 | -25.68 | 1 | 14.85 |
| Total Runoff | Upper Mississippi | 1 | -47.18 | 3 | -36.01 |
| Soil Moisture | Arkansas-Red-White | 3 | -14.46 | 1 | -5.79 |
| Soil Moisture | Lower Mississippi | 1 | -23.57 | 0 | -19.68 |
| Soil Moisture | Missouri | 1 | -14.56 | 2 | 16.30 |
| Soil Moisture | Ohio | 2 | -29.20 | 2 | -21.45 |
| Soil Moisture | Upper Mississippi | 1 | -10.65 | 1 | -5.47 |
| Snowmelt | Arkansas-Red-White | 0 | 49.91 | 1 | 0 |

| | | | | | |
|---|---|---|---|---|---|
| Snowmelt | Lower Mississippi | 0 | 19.46 | 4 | 0 |
| Snowmelt | Missouri | 1 | 23.46 | 0 | 0 |
| Snowmelt | Ohio | 0 | -20.62 | 2 | 0 |
| Snowmelt | Upper Mississippi | 0 | -5.20 | 1 | 0 |
| Evapotranspiration | Arkansas-Red-White | 1 | 25.00 | 0.00 | 14.29 |
| Evapotranspiration | Lower Mississippi | 1 | 2.08 | 0.00 | -16.67 |
| Evapotranspiration | Missouri | 1 | 36.36 | 2.00 | -25.00 |
| Evapotranspiration | Ohio | 1 | -8.16 | 0.00 | -14.29 |
| Evapotranspiration | Upper Mississippi | 1 | -2.08 | 2.00 | -40.00 |

 **Table A2: Lag Correlation values for each variable in the Eastern, Western, and Entire Mississippi Basin at each lag. Maximum correlation values are bolded.**

| Lag | Basin | Temperature | Precipitation | Surface Runoff | Subsurface Runoff | Total Runoff | Soil Moisture | Snow Melt | Evapotranspiration |
|---|---|---|---|---|---|---|---|---|---|
| -11 | Eastern | -0.445 | 0.071 | 0.103 | 0.126 | 0.125 | 0.365 | 0.427 | -0.367 |
| -10 | Eastern | -0.606 | -0.036 | 0.299 | 0.39 | 0.387 | 0.547 | 0.709 | -0.49 |
| -9 | Eastern | -0.676 | 0.252 | 0.586 | 0.653 | 0.661 | 0.687 | 0.605 | -0.529 |
| -8 | Eastern | -0.692 | 0.259 | 0.669 | 0.791 | 0.789 | 0.732 | 0.038 | -0.504 |
| -7 | Eastern | -0.653 | 0.436 | **0.714** | 0.803 | 0.822 | 0.675 | -0.261 | -0.425 |
| -6 | Eastern | -0.538 | 0.477 | 0.667 | 0.717 | 0.794 | 0.524 | -0.313 | -0.289 |
| -5 | Eastern | -0.333 | **0.648** | 0.615 | 0.613 | 0.748 | 0.352 | -0.315 | -0.079 |
| -4 | Eastern | -0.031 | 0.577 | 0.51 | 0.633 | 0.723 | 0.159 | -0.231 | 0.226 |
| -3 | Eastern | 0.347 | 0.517 | 0.435 | 0.776 | 0.78 | -0.026 | 0.283 | 0.589 |
| -2 | Eastern | 0.729 | 0.263 | 0.265 | 0.919 | **0.841** | -0.159 | 0.586 | 0.884 |
| -1 | Eastern | **0.968** | 0.314 | 0.193 | **0.939** | 0.828 | -0.083 | 0.843 | **0.978** |
| 0 | Eastern | 0.945 | 0.241 | 0.078 | 0.748 | 0.675 | **0.779** | **0.947** | 0.803 |
| 1 | Eastern | 0.714 | -0.227 | -0.317 | 0.278 | 0.133 | -0.267 | 0.489 | 0.442 |
| 2 | Eastern | 0.425 | -0.216 | -0.545 | -0.23 | -0.371 | -0.538 | -0.087 | 0.076 |
| 3 | Eastern | 0.182 | -0.377 | -0.733 | -0.639 | -0.747 | -0.721 | -0.457 | -0.186 |
| 4 | Eastern | -0.005 | -0.41 | -0.766 | -0.871 | -0.926 | -0.789 | -0.525 | -0.343 |
| 5 | Eastern | -0.146 | -0.548 | -0.759 | -0.917 | -0.935 | -0.733 | -0.526 | -0.432 |
| 6 | Eastern | -0.253 | -0.543 | -0.61 | -0.811 | -0.814 | -0.564 | -0.519 | -0.477 |
| 7 | Eastern | -0.33 | -0.634 | -0.409 | -0.644 | -0.646 | -0.361 | -0.469 | -0.502 |
| 8 | Eastern | -0.37 | -0.51 | -0.196 | -0.503 | -0.497 | -0.129 | -0.304 | -0.524 |
| 9 | Eastern | -0.388 | -0.414 | -0.075 | -0.38 | -0.37 | 0.106 | -0.058 | -0.531 |
| 10 | Eastern | -0.389 | 0.023 | 0.152 | -0.244 | -0.22 | 0.302 | 0.111 | -0.473 |
| 11 | Eastern | -0.374 | 0.105 | 0.177 | -0.115 | -0.077 | 0.308 | 0.194 | -0.358 |
| -11 | Western | -0.44 | -0.403 | -0.359 | -0.19 | -0.269 | 0.221 | 0.037 | -0.329 |
| -10 | Western | -0.533 | -0.578 | -0.465 | -0.233 | -0.34 | 0.415 | 0.297 | -0.443 |
| -9 | Western | -0.617 | -0.609 | -0.437 | -0.195 | -0.308 | 0.617 | 0.765 | -0.488 |
| -8 | Western | -0.674 | -0.572 | -0.363 | -0.069 | -0.212 | 0.748 | 0.696 | -0.495 |
| -7 | Western | -0.679 | -0.462 | -0.204 | 0.114 | -0.052 | **0.798** | 0.315 | -0.456 |
| -6 | Western | -0.606 | -0.335 | -0.03 | 0.37 | 0.154 | 0.741 | -0.05 | -0.361 |
| -5 | Western | -0.421 | -0.182 | 0.118 | 0.484 | 0.296 | 0.515 | -0.381 | -0.177 |
| -4 | Western | -0.099 | 0.079 | 0.32 | 0.566 | 0.459 | 0.279 | -0.335 | 0.132 |
| -3 | Western | 0.34 | 0.424 | 0.569 | 0.648 | 0.651 | 0.129 | -0.012 | 0.529 |
| -2 | Western | 0.764 | 0.751 | 0.767 | 0.706 | 0.797 | 0.039 | 0.381 | 0.866 |
| -1 | Western | **0.981** | **0.942** | **0.846** | **0.719** | **0.832** | 0.073 | 0.807 | **0.99** |
| 0 | Western | 0.937 | 0.883 | 0.703 | 0.635 | 0.656 | 0.696 | **0.957** | 0.834 |
| 1 | Western | 0.662 | 0.612 | 0.406 | 0.202 | 0.282 | -0.153 | 0.726 | 0.466 |
| 2 | Western | 0.297 | 0.324 | 0.105 | -0.084 | -0.031 | -0.418 | 0.271 | 0.062 |

| | | | | | | | | |
|---|---|---|---|---|---|---|---|---|
| 3 | Western | -0.026 | 0.071 | -0.134 | -0.255 | -0.249 | -0.646 | -0.291 | -0.235 |
| 4 | Western | -0.275 | -0.094 | -0.284 | -0.378 | -0.387 | -0.779 | -0.649 | -0.395 |
| 5 | Western | -0.458 | -0.243 | -0.407 | -0.481 | -0.492 | -0.819 | -0.762 | -0.458 |
| 6 | Western | -0.572 | -0.345 | -0.502 | -0.63 | -0.595 | -0.751 | -0.685 | -0.48 |
| 7 | Western | -0.573 | -0.389 | -0.502 | -0.681 | -0.602 | -0.51 | -0.481 | -0.488 |
| 8 | Western | -0.361 | -0.408 | -0.47 | -0.635 | -0.557 | -0.239 | -0.286 | -0.501 |
| 9 | Western | 0.146 | -0.439 | -0.46 | -0.529 | -0.502 | -0.036 | -0.163 | -0.525 |
| 10 | Western | 0.428 | -0.432 | -0.403 | -0.391 | -0.403 | 0.109 | -0.084 | -0.472 |
| 11 | Western | 0.371 | -0.354 | -0.32 | -0.244 | -0.283 | 0.131 | -0.054 | -0.338 |
| -11 | Entire | -0.441 | -0.385 | -0.304 | 0.009 | -0.076 | 0.258 | 0.116 | -0.34 |
| -10 | Entire | -0.562 | -0.566 | -0.341 | 0.171 | 0.042 | 0.452 | 0.401 | -0.465 |
| -9 | Entire | -0.637 | -0.561 | -0.227 | 0.371 | 0.221 | 0.642 | 0.794 | -0.513 |
| -8 | Entire | -0.681 | -0.512 | -0.108 | 0.555 | 0.378 | 0.755 | 0.621 | -0.488 |
| -7 | Entire | -0.675 | -0.369 | 0.087 | 0.689 | 0.531 | **0.781** | 0.22 | -0.389 |
| -6 | Entire | -0.591 | -0.231 | 0.255 | 0.762 | 0.663 | 0.702 | -0.113 | -0.233 |
| -5 | Entire | -0.4 | -0.07 | 0.358 | 0.708 | 0.712 | 0.484 | -0.391 | -0.05 |
| -4 | Entire | -0.082 | 0.17 | 0.483 | 0.69 | 0.749 | 0.255 | -0.278 | 0.107 |
| -3 | Entire | 0.343 | 0.473 | 0.648 | 0.757 | 0.811 | 0.094 | 0.084 | 0.306 |
| -2 | Entire | 0.761 | 0.739 | 0.726 | 0.841 | **0.832** | -0.008 | 0.464 | 0.564 |
| -1 | Entire | **0.982** | **0.909** | **0.729** | **0.873** | 0.768 | 0.035 | 0.837 | 0.807 |
| 0 | Entire | 0.939 | 0.854 | 0.537 | 0.793 | 0.551 | 0.729 | **0.966** | **0.934** |
| 1 | Entire | 0.675 | 0.568 | 0.23 | 0.332 | 0.082 | -0.18 | 0.695 | 0.783 |
| 2 | Entire | 0.326 | 0.317 | -0.039 | -0.154 | -0.316 | -0.451 | 0.22 | 0.492 |
| 3 | Entire | 0.02 | 0.079 | -0.268 | -0.559 | -0.609 | -0.672 | -0.331 | 0.179 |
| 4 | Entire | -0.216 | -0.064 | -0.398 | -0.817 | -0.772 | -0.791 | -0.666 | -0.086 |
| 5 | Entire | -0.39 | -0.223 | -0.521 | -0.92 | -0.845 | -0.809 | -0.781 | -0.296 |
| 6 | Entire | -0.504 | -0.338 | -0.604 | -0.897 | -0.848 | -0.717 | -0.716 | -0.444 |
| 7 | Entire | -0.536 | -0.407 | -0.561 | -0.786 | -0.762 | -0.482 | -0.508 | -0.51 |
| 8 | Entire | -0.42 | -0.428 | -0.475 | -0.661 | -0.648 | -0.216 | -0.287 | -0.519 |
| 9 | Entire | 0.041 | -0.46 | -0.437 | -0.526 | -0.537 | -0.003 | -0.143 | -0.527 |
| 10 | Entire | 0.429 | -0.409 | -0.317 | -0.364 | -0.381 | 0.156 | -0.051 | -0.475 |
| 11 | Entire | 0.371 | -0.328 | -0.241 | -0.192 | -0.215 | 0.175 | -0.008 | -0.344 |

 **Table A3: Spectral Angle values for each variable in the Eastern, Western and Entire Mississippi Basin**

| Basin | Variable | Spectral Angle |
|---|---|---|
| Eastern Mississippi Basin | Precipitation | 0.388 |
| Western Mississippi Basin | Precipitation | 0.378 |
| Entire Mississippi Basin | Precipitation | 0.383 |
| Eastern Mississippi Basin | Subsurface Runoff | 0.64 |
| Western Mississippi Basin | Subsurface Runoff | 0.628 |
| Entire Mississippi Basin | Subsurface Runoff | 0.543 |
| Eastern Mississippi Basin | Surface Runoff | 0.681 |
| Western Mississippi Basin | Surface Runoff | 0.628 |
| Entire Mississippi Basin | Surface Runoff | 0.576 |
| Eastern Mississippi Basin | Total Runoff | 0.604 |
| Western Mississippi Basin | Total Runoff | 0.578 |
| Entire Mississippi Basin | Total Runoff | 0.545 |
| Eastern Mississippi Basin | Temperature | 0.266 |
| Western Mississippi Basin | Temperature | 0.391 |
| Entire Mississippi Basin | Temperature | 0.354 |
| Eastern Mississippi Basin | Soil Moisture | 0.098 |
| Western Mississippi Basin | Soil Moisture | 0.131 |
| Entire Mississippi Basin | Soil Moisture | 0.115 |
| Eastern Mississippi Basin | Evapotranspiration | 0.384 |
| Western Mississippi Basin | Evapotranspiration | 0.411 |
| Entire Mississippi Basin | Evapotranspiration | 0.256 |
| Eastern Mississippi Basin | Snowmelt | 0.733 |
| Western Mississippi Basin | Snowmelt | 0.484 |
| Entire Mississippi Basin | Snowmelt | 0.474 |

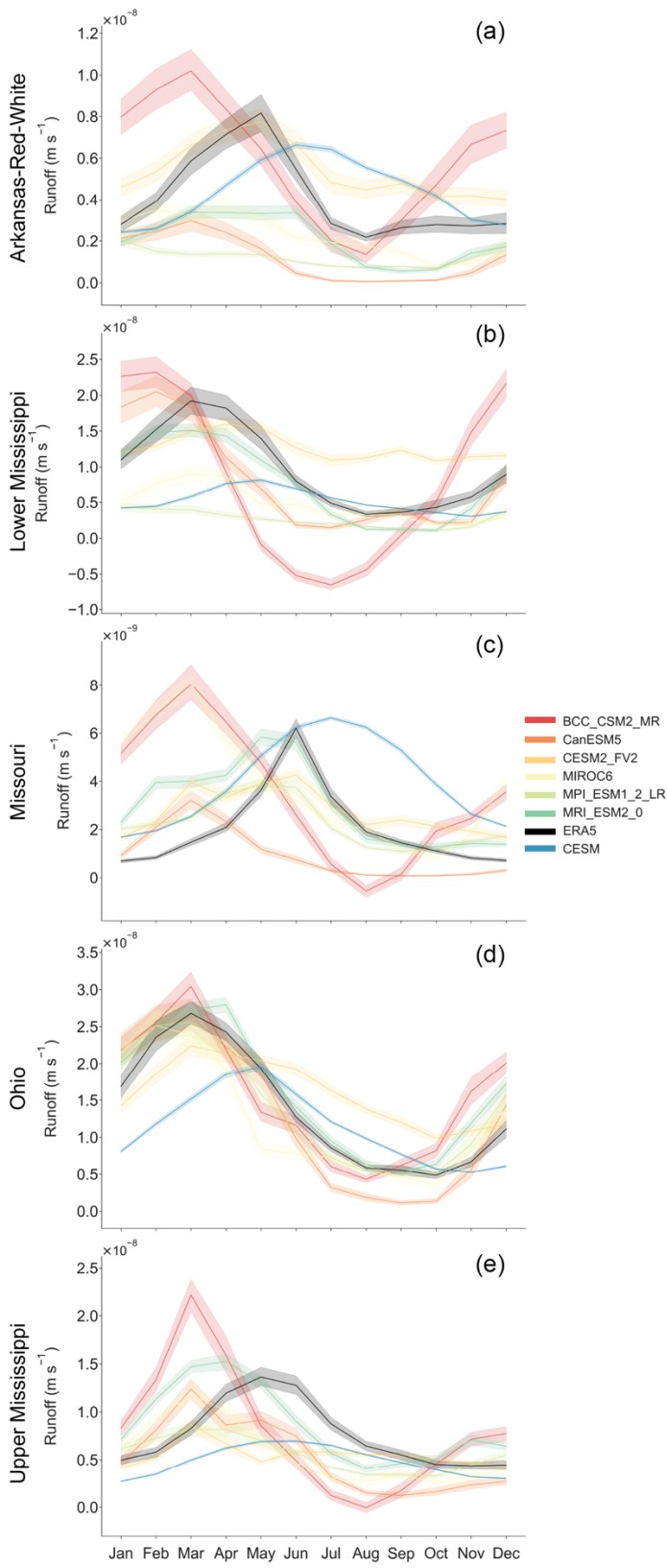

**Figure A4: Monthly mean simulated runoff (m/s) from selected CMIP6 models (BCC CSM2 MR [red], CanESM5 [dark orange], CEMS2 [light orange], MIROC6 [yellow], MPI ESM1 2 LR [light green], MRI ESM2 0 [dark green]), CESM1 [blue], and ERA5 reanalysis [black] for the Arkansas-Red-White (a), Lower Mississippi (b), Missouri (c), Ohio (d), and Upper Mississippi (e) basins. Shading represents a 95% confidence interval from interannual variability.**