# Peer review of "Evaluation of hydroclimatic biases in the Community Earth System Model (CESM1) within the Mississippi River basin"

_Hydrology and Earth System Sciences, 2024_

## Author Comment (AC1)

**Author Comment: Evaluation of hydroclimatic biases in the Community Earth System Model (CESM1) within the Mississippi River Basin**

*Reviewer 1*

In this paper, O'Donnell evaluated hydroclimatic biases in the CESM1 within the Mississippi River basin. The evaluation data include USGS gauge data of river discharge, ERA5 reanalysis, GPCC precipitation observations, and LIvneh ET. They also compared the CESM1 simulated runoff with the simulations from several CMIP6 models, including the newer version of CESM - CESM2. They demonstrated that CESM1 has substantial biases in simulating runoff and river discharge and attributed the model discrepancy to the deficiency in the RTM river model. They showed that CESM2 with the more advanced MOSART river model performs better in the river basin.

We thank Reviewer 1 for the thoughtful and useful comments on our manuscript. We do agree with many of these comments and will adopt these changes to better represent the hydroclimatic biases in CESM1 in the Mississippi River Basin and motivations of the work.

While the results are clearly presented, I find that the motivations of this study are not clear and there are likely serious errors in the CEMS1 configuration or simulation or both. As such, the study has limited values and I cannot recommend its publication in this journal. There are two major gaps/issues in the paper. First, the authors have not explained clearly why we need to know the biases of the old CESM1 given that the newer version CESM2 has been used by CMIP6. Does CESM1 have unique features that are not available in CESM2? Is there still a large user base who is using these features for important studies? What are the obstacles that hinder the users to adapt to the new version? Without good reasons, I would question why not to evaluate CESM2 instead.

We agree with Reviewer 1 that the motivations can be further clarified. CESM1 is still of significant value because it is one of the few CMIP5 models that has both a routing model and multiple available modeling projects, including the Last Millennium Ensemble (CESM-LME) (Otto-Bliesner et al., 2016), which includes both full-forcing and single-forcing simulations for the period 850-2005 CE. While we demonstrate here that CESM2, which is a part of the CMIP6 suite and uses MOSART, has significantly improved seasonal timing, neither CESM2 nor other CMIP6 GCMs yet include equivalents of the LME project simulations. Moreover, it is useful to evaluate the degree to which CESM2/MOSART represents an improvement over CESM1/RTM – which we do. A number of studies still use CESM1 to investigate hydroclimate over the last millennium. For example, recently published papers such as *PDO influenced interdecadal summer*

*precipitation change over East China in mid-18th century* (Chen et al. 2024. *Nature Climate and Atmospheric Science* 7 (1): 1–11. https://doi.org/10.1038/s41612-024-00666-6.) and *Influence of ENSO and Volcanic Eruptions on Himalayan Jet Latitude* (Thapa, Uday Kunwar, and Samantha Stevenson. 2024. *Geophysical Research Letters* 51 (14): e2023GL107271. https://doi.org/10.1029/2023GL107271.), among others rely on CESM1. Any studies using CESM1 must take into consideration the biases in CESM1, particularly if they focus on the Mississippi River Basin, and if a study focuses on other regions, similar biases should be evaluated before conclusions on hydroclimatic changes are drawn.

Second, the model simulations look suspicious. Table 3 indicates that the modeled surface runoff, subsurface runoff, total runoff and snowmelt are two orders of magnitude smaller than the observations or benchmark data. Given this unbelievably poor performance, I would honestly think the model is useless. It is reasonable to question whether the authors have configured the model or extracted the outputs correctly because CESM1 has been well tested before. Furthermore, there are also several other variables with odd values: 1) the reported precipitation values (Figure 1b and Line 89) are less than 200 mm/year which if true would mean that the Mississippi River basin would be a desert; and 2) the reported runoff values in Figures 3 and the reported snowmelt values in Figure 4 if converted to mm/yr are unrealistically large (1e-6 m/s > 3e4 mm/yr). Also, it is very strange that the model and data are not shown at the same scales in the model-data comparisons (Figures 3, 4, and 7).

We agree with Reviewer 1 that the differences in magnitudes between simulations and observations are large. We have checked this previously, but will confirm any conversions and all values again. We can also add a note in any figure captions where different y-axis scaling is used. While CESM1 has been widely used and some diagnostics are available, the user documentation specifies the need for rigorous validation ("CESM Overview." n.d. CEMS1.1.z User's Guide. Accessed August 9, 2024. https://www2.cesm.ucar.edu/models/cesm1.1/cesm/doc/usersguide/x21.html).

We also thank Reviewer 1 for pointing out this error in reported precipitation value units. The units should be mm/month, and this will be corrected in an updated manuscript.

**My other comments are shown below.**

L104: Why did you choose the USGS 07289000 which is only available since 2008 for the Lower MS? Why not choose 07295100 Mississippi River at Tarbert Landing, Mississippi which has much longer data records for investigation?

We agree with Reviewer 1 that longer records are useful for comparison. The USGS 07295100 Mississippi River at Tarbert Landing does not have monthly discharge data, but the US Army Corps does for this site. The USGS 07289000 gage data was used for the comparison of simulated to observed data from gages, but the 07295100 Mississippi River at Tarbert Landing data was used in the pre- and post-dam comparison. In an updated manuscript, we will update Tables 1 & 2, and Figure 2 to reflect data incorporated from 07295100 Mississippi River at Tarbert Landing and update citations accordingly.

Table 2: Could you explain why these years can be regarded as the separation of pre-modification vs. post-modification? For example, for the Missouri River, many of the dams were constructed in the 1930s. As a result, I do not think you can see much difference by comparing the model simulation before and after 1967.

We agree with Reviewer 1 that the delineation between pre- and post- modification years needs to be well supported. The years listed in Table 2 are the end of the periods of major river modification when the impacts of the modifications are fully implemented, based on the literature cited in the Table 2 caption. For example, Jacobson & Galat (2008) note that "The six mainstem reservoirs were constructed between 1937 and 1963 and operation as a system began in 1967" on the Missouri. We can further clarify the choice of these years for the subbasins in a revised manuscript.

Section 2: Given the importance of runoff generation and river routing in this study, wouldn't it be necessary to describe CLM and RTM briefly? Particularly, how is water management represented in RTM?

We thank Reviewer 1 for pointing this out, descriptions of CLM, RTM, and representation of water management are useful context and we can add these in a revised manuscript.

L135: QOVER is only a part of surface runoff and does not include surface runoff from standing water (QH2OSFC).

We agree that it is important to include all components of surface runoff. However, in the CESM1 LME project from which we are retrieving data, standing water (QH2OSFC) is not available as a variable and QOVER is representative of surface runoff. While RTM uses CLM4, we also found that QOVER is noted as including QHSOSFC in CLM5 ("Questions on Runoff Components in CLM5 BGC-CROP Mode (Ctsm5.1.Dev118)." 2024. *DiscussCESM Forums*. https://bb.cgd.ucar.edu/cesm/threads/questions-on-runoff-components-in-clm5-bgc-crop-mode-ctsm5-1-dev118.9271/.). We did not include the additionally available surface runoff term, QRGWL (surface runoff at glaciers (liquid

only) wetland lakes), since this project does not investigate runoff or the hydrologic cycle components in a glaciated area.

L144: Could you describe briefly these 13 ensemble members? Under what configurations these members were simulated?

We agree with Reviewer 1 that a description of the 13 ensemble members from the CESM LME project, as well as their configurations, are important context and we will add this description in section 2.4 Earth system models and validation approach, where the ensemble members are introduced.

L151: What software do you use to calculate lagged correlation and spectral angle?

We agree that this should be specified and will update the methods section of the manuscript to include that lagged correlation and spectral angle were calculated in Python with the pandas corr and HydroErr sa functions, respectively.

L265: It is probably not true. To my understanding, RTM does not represent two-way land-river coupling. As such, subsurface runoff affects river routing but not vice versa.

We thank Reviewer 1 for pointing out that clarification is needed here. We agree that RTM does not represent two-way land-river coupling. In this paragraph, we intended to highlight that RTM has been shown to have a time lag between runoff and discharge, not suggest that the delayed seasonality of discharge is contributing to any of the timing offsets in runoff. We will edit this paragraph for clarity in an updated manuscript.

Section 3: Please separate results and discussion. The current structure prevents a cohesive storytelling.

We agree with the reviewer that in many cases it is useful to separate the results and discussion. Because of the number of variables being examined, we found it was clearer to group the results and discussion by variables and skill metrics, rather than presenting them as separate sections.

Section 3.3: Why isn't this metric introduced in the methods?

We thank Reviewer 1 for pointing out that clarification is needed. Section 3.3 discusses relative difference, which is introduced in section 2.4 of the Methods and Data section, starting on line 154. We will clarify this section of the Methods in an updated manuscript.

---

## Author Comment (AC2)

**Author Comment: Evaluation of hydroclimatic biases in the Community Earth System Model (CESM1) within the Mississippi River Basin**

*Reviewer 2*

The authors investigated the biases in monthly atmospheric and land surface variables in CESM1 data in the Mississippi River Basin (MRB). They found that there exist large seasonal biases in CESM1 precipitation, runoff, and discharge simulations. By comparing with other CMIP6 model data, they showed that the CESM2 model had better precipitation seasonality than CESM1 model. While this study provides useful information about the CESM1 model quality in the MRB, a few major issues need to be addressed before this paper can be accepted.

We thank Reviewer 2 for the thoughtful comments on our manuscript.

Major comments:

1. The primary concern is the value of evaluating only the CESM1 model when CESM2 large ensemble project data are available for this region. I suggest reorganizing this study to evaluate both CESM1 and CESM2 data and highlight the improvements in the CESM2 model.

   We thank Reviewer 2 for this suggestion and agree that emphasizing the improvements in CESM2 is important. However, we believe that addressing Reviewer 1's comments on clarifying the motivations will better address this. By more clearly stating the value of CESM1 – it being one of the few CMIP5 models that has both a routing model and multiple available modeling projects, including the Last Millennium Ensemble (CESM-LME), which is still widely in use – we can then focus attention on the relevant hydrologic biases present in the region that impact the seasonality of discharge for those that still rely on the CESM1 model output for lack of other options.

2. The division of the MRB into western and eastern parts may not be sufficiently justified. The major sub-basins in the western part, i.e., Upper Mississippi, Missouri, and Arkansas River Basins, have very different atmospheric and hydrological properties. I recommend conducting comparison individually for each major river basins.

   We agree with Reviewer 2 that the major subbasins of the Mississippi River basin have a range of atmospheric and hydrologic properties. We began our investigation by comparing each major subbasin individually rather than by grouping them into the eastern and western parts, but found the seasonal trends among individual basins to be similar enough that grouping them streamlined the results. We can add results from the major sub-basins to an appendix.

3. Figures 3, 4, 7: These figures have two y-axes with different ranges on each side. Does this indicate that the climate model data have significantly different magnitudes compared to reanalysis/observations? I suggest using consistent y-axis ranges for these figures to avoid misleading interpretations, but I understand that the authors may want to compare seasonality patterns rather than magnitudes.

We agree with Reviewer 2 that one y-axis is the most ideal plot layout. With the difference in magnitude between the data sets, the seasonality patterns are not visible if they are plotted on one y-axis; we will add a note in any figure captions where different y-axis scaling is used to avoid misleading interpretations.

4. Line 110: The gauge record was divided into pre- and post-modification periods. Please provide references to support the chosen separation year. Also, I suggest a more thorough discussion of the influences of human modification on river discharge in the MRB. The main stem of the river was heavily controlled by dams and flood protection structures. The influence of these structures on river flows should be mentioned and highlighted.

We agree with Reviewer 2 that the delineation between pre- and post- modification years needs to be well supported. The years listed in Table 2 are the end of the periods of major river modification when the impacts of the modifications are fully implemented, based on the literature cited in the Table 2 caption. We can further clarify the choice of these years in the text in a revised manuscript, along with more discussion on the influence of dams and flood control structures in the basin.

Minor comments:
1. Line 150: Please provide more details about the lag correlation and spectral angle methods. What formulas were used to calculate these two metrics?

We agree with Reviewer 2 that this should be specified, and will update the methods section of the manuscript to include this.

2. Table 3: If possible, consider using figures instead of tables to present these results.

We thank the reviewer for pointing out that clarification is needed. The results in Table 3 summarize values from Figures 2,3, and 4 that are discussed in the text, so we will add clarification in the captions and text or rearrange content for clarity.

3. Figure 2: Adding the names of gauges used for each sub-basin would be helpful.

We agree with Reviewer 2 and can add the names of the gages in addition to the numbers.

4. Figure 7: Please specify the uncertainty range in the figure caption. Is it a 95% uncertainty interval? Also, clarify the source of uncertainty. Is it coming from inter-annual variability or ensemble variability?

We agree with the reviewer that this should be specified and will update Figure 7 accordingly.

---

## Author Response (AR1)

**Author Comment: Evaluation of hydroclimatic biases in the Community Earth System Model (CESM1) within the Mississippi River Basin**

*Reviewer 1*

In this paper, O'Donnell evaluated hydroclimatic biases in the CESM1 within the Mississippi River basin. The evaluation data include USGS gauge data of river discharge, ERA5 reanalysis, GPCC precipitation observations, and LIvneh ET. They also compared the CESM1 simulated runoff with the simulations from several CMIP6 models, including the newer version of CESM - CESM2. They demonstrated that CESM1 has substantial biases in simulating runoff and river discharge and attributed the model discrepancy to the deficiency in the RTM river model. They showed that CESM2 with the more advanced MOSART river model performs better in the river basin.

We thank Reviewer 1 for the thoughtful and useful comments on our manuscript. We agree with many of these comments and adopted these changes to better represent the hydroclimatic biases in CESM1 in the Mississippi River Basin and motivations of the work.

While the results are clearly presented, I find that the motivations of this study are not clear and there are likely serious errors in the CEMS1 configuration or simulation or both. As such, the study has limited values and I cannot recommend its publication in this journal. There are two major gaps/issues in the paper. First, the authors have not explained clearly why we need to know the biases of the old CESM1 given that the newer version CESM2 has been used by CMIP6. Does CESM1 have unique features that are not available in CESM2? Is there still a large user base who is using these features for important studies? What are the obstacles that hinder the users to adapt to the new version? Without good reasons, I would question why not to evaluate CESM2 instead.

We agree with Reviewer 1 that the motivations can be further clarified. CESM1 is still of significant value because it is one of the few CMIP5 models that has both a routing model and multiple available modeling projects, including the Last Millennium Ensemble (CESM-LME) (Otto-Bliesner et al., 2016), which includes both full-forcing and single-forcing simulations for the period 850-2005 CE. While we demonstrate here that CESM2, which is a part of the CMIP6 suite and uses MOSART, has significantly improved seasonal timing, neither CESM2 nor other CMIP6 GCMs yet include equivalents of the LME project simulations. Moreover, it is useful to evaluate the degree to which CESM2/MOSART represents an improvement over CESM1/RTM – which we do. A number of studies still use CESM1 to investigate hydroclimate over the last millennium. For example, recently published papers such as *PDO influenced interdecadal summer precipitation change over East China in mid-18th century* (Chen et al. 2024. *Nature Climate and Atmospheric Science* 7 (1): 1–11. https://doi.org/10.1038/s41612-024-00666-6.) and *Influence of ENSO and Volcanic Eruptions on Himalayan Jet Latitude* (Thapa, Uday Kunwar, and Samantha Stevenson. 2024. *Geophysical Research Letters* 51 (14): e2023GL107271. https://doi.org/10.1029/2023GL107271.), among others rely on CESM1. Any studies using CESM1 must take into consideration the biases in CESM1, particularly if they focus on the Mississippi River Basin, and if a study focuses on other regions, similar biases should be evaluated before conclusions on hydroclimatic changes are drawn.

To clarify this in the manuscript, we updated the fourth paragraph of the introduction (lines 79-88 in the track-changes file; lines 63-72 in the final document).

Second, the model simulations look suspicious. Table 3 indicates that the modeled surface runoff, subsurface runoff, total runoff and snowmelt are two orders of magnitude smaller than the observations or benchmark data. Given this unbelievably poor performance, I would honestly think the model is useless. It is reasonable to question whether the authors have configured the model or extracted the outputs correctly because CESM1 has been well tested before. Furthermore, there are also several other variables with odd values: 1) the reported precipitation values (Figure 1b and Line 89) are less than 200 mm/year which if true would mean that the Mississippi River basin would be a desert; and 2) the reported runoff values in Figures 3 and the reported snowmelt values in Figure 4 if converted to mm/yr are unrealistically large (1e-6 m/s > 3e4 mm/yr). Also, it is very strange that the model and data are not shown at the same scales in the model-data comparisons (Figures 3, 4, and 7).

We thank Reviewer 1 again for bringing this back to our attention.  Upon examining this closely again, we did find an error in the conversion, which explains some of the discrepancy between the values. We have now updated these values in all relevant figures (3,4,7), and in the calculation of relative difference (Tables 3,4,5) and values referenced throughout. Since this a linear conversion factor, it does not change the timing of minimum or maximum values in the data being compared, and does not alter the major conclusions of the work.

While CESM1 has been widely used and some diagnostics are available, the user documentation specifies the need for rigorous validation ("CESM Overview." n.d. CEMS1.1.z User's Guide. Accessed August 9, 2024. https://www2.cesm.ucar.edu/models/cesm1.1/cesm/doc/usersguide/x21.html). Since the differences in seasonality are not impacted by conversion factors, identifying causes of seasonal shifts in modeled flow in this major river basin are still important to explore.

We also thank Reviewer 1 for pointing out this error in reported precipitation value units. The units should be mm/month, not mm/year, and were updated in the caption for Figure 1 and Section 2.1 (lines 108-110 in the track-changes file; lines 92-95 in the final document).

**My other comments are shown below.**

L104: Why did you choose the USGS 07289000 which is only available since 2008 for the Lower MS? Why not choose 07295100 Mississippi River at Tarbert Landing, Mississippi which has much longer data records for investigation?

We agree with Reviewer 1 that longer records are useful for comparison. We updated Figures 1 and 2 and Tables 1 and 2 to use the USACE Mississippi River at Tarbert Landing gage data. The Tarbert Landing data was already used in the pre- and post-dam comparison (Appendix Figure 1). Citations and references were updated throughout the text.

Table 2: Could you explain why these years can be regarded as the separation of pre-modification vs. post-modification? For example, for the Missouri River, many of the dams were constructed in the 1930s. As a result, I do not think you can see much difference by comparing the model simulation before and after 1967.

We agree with Reviewer 1 that the delineation between pre- and post- modification years needs to be well supported. The years listed in Table 2 are the end of the periods of major river modification when the impacts of the modifications are fully implemented, based on the literature cited in the Table 2 caption. For example, Jacobson & Galat (2008) note that "The six mainstem reservoirs were constructed between 1937 and 1963 and operation as a system began in 1967" on the Missouri. We added text to further clarify the choice of these years for the subbasins in Section 2.2 (lines 148-179 in the track-changes file; lines 122-129 in the final document).

Section 2: Given the importance of runoff generation and river routing in this study, wouldn't it be necessary to describe CLM and RTM briefly? Particularly, how is water management represented in RTM?

We thank Reviewer 1 for pointing this out, descriptions of CLM, RTM, and representation of water management are useful context and we added descriptions of these in Section 2.4 (lines 234-244 in the track-changed file; lines 162-172 in the final document).

L135: QOVER is only a part of surface runoff and does not include surface runoff from standing water (QH2OSFC).

We agree that it is important to include all components of surface runoff. However, in the CESM1 LME project from which we are retrieving data, standing water (QH2OSFC) is not available as a variable and QOVER is representative of surface runoff. While RTM uses CLM4, we also found that QOVER is noted as including QHSOSFC in CLM5 ("Questions on Runoff Components in CLM5 BGC-CROP Mode (Ctsm5.1.Dev118)." 2024. *DiscussCESM Forums*. https://bb.cgd.ucar.edu/cesm/threads/questions-on-runoff-components-in-clm5-bgc-crop-mode-ctsm5-1-dev118.9271/.). We did not include the additionally available surface runoff term, QRGWL (surface runoff at glaciers (liquid only) wetland lakes), since this project does not investigate runoff or the hydrologic cycle components in a glaciated area. We noted this in Section 2.4 (line 224-226 in the track-changed file; lines 152-154 of the final document).

L144: Could you describe briefly these 13 ensemble members? Under what configurations these members were simulated?

We agree with Reviewer 1 that a description of the 13 ensemble members is important context, and this was updated in section 2.4 (lines 228-232 in the track-changed document; lines 156-160 in the final document).

L151: What software do you use to calculate lagged correlation and spectral angle?

We agree that this should be specified and was updated the methods section in lines 260-261 in the track-changed file and in lines 187-189 of the final document.

L265: It is probably not true. To my understanding, RTM does not represent two-way land-river coupling. As such, subsurface runoff affects river routing but not vice versa.

We thank Reviewer 1 for pointing out that clarification is needed here. We agree that RTM does not represent two-way land-river coupling. In this paragraph, we intended to highlight that RTM has been shown to have a time lag between runoff and discharge, not suggest that the delayed seasonality of discharge is contributing to any of the timing offsets in runoff. We edited this paragraph for clarity in lines 440-441 in the track-changed document (lines 324-325 in the final document).

Section 3: Please separate results and discussion. The current structure prevents a cohesive storytelling.

We agree with the reviewer that in many cases it is useful to separate the results and discussion. Because of the number of variables being examined, we found it was clearer to group the results and discussion by variables and skill metrics, rather than presenting them as separate sections.

Section 3.3: Why isn't this metric introduced in the methods?

We thank Reviewer 1 for pointing out that clarification is needed. Section 3.3 discusses relative difference, which is introduced in section 2.4 of the Methods and Data section. We updated the Methods section in the track-changed manuscript to include more information on relative difference from lines 264-270 (lines 191-197 in the final document).

**Author Comment: Evaluation of hydroclimatic biases in the Community Earth System Model (CESM1) within the Mississippi River Basin**

*Reviewer 2*

The authors investigated the biases in monthly atmospheric and land surface variables in CESM1 data in the Mississippi River Basin (MRB). They found that there exist large seasonal biases in CESM1 precipitation, runoff, and discharge simulations. By comparing with other CMIP6 model data, they showed that the CESM2 model had better precipitation seasonality than CESM1 model. While this study provides useful information about the CESM1 model quality in the MRB, a few major issues need to be addressed before this paper can be accepted.

We thank Reviewer 2 for the thoughtful comments on our manuscript.

Major comments:
1. The primary concern is the value of evaluating only the CESM1 model when CESM2 large ensemble project data are available for this region. I suggest reorganizing this study to evaluate both CESM1 and CESM2 data and highlight the improvements in the CESM2 model.

We thank Reviewer 2 for this suggestion and agree that emphasizing the improvements in CESM2 is important. However, we believe that addressing Reviewer 1's comments on clarifying the motivations will better address this. To clarify these in the manuscript, we updated the fourth paragraph of the introduction (lines 79-88 in the track-changes file; lines 63-72 in the final document).

By more clearly stating the value of CESM1 – it being one of the few CMIP5 models that has both a routing model and multiple available modeling projects, including the Last Millennium Ensemble (CESM-LME), which is still widely in use – we can then focus attention on the relevant hydrologic biases present in the region that impact the seasonality of discharge for those that still rely on the CESM1 model output for lack of other options.

2. The division of the MRB into western and eastern parts may not be sufficiently justified. The major sub-basins in the western part, i.e., Upper Mississippi, Missouri, and Arkansas River Basins, have very different atmospheric and hydrological properties. I recommend conducting comparison individually for each major river basins.

We agree with Reviewer 2 that the major subbasins of the Mississippi River basin have a range of atmospheric and hydrologic properties. We began our investigation by comparing each major subbasin individually rather than by grouping them into the eastern and western parts, but found the seasonal trends among individual basins to be similar enough that grouping them streamlined the results. We added results from the major sub-basins to Appendix Figures 2,3, and 4 and Appendix Table 1.

3. Figures 3, 4, 7: These figures have two y-axes with different ranges on each side. Does this indicate that the climate model data have significantly different magnitudes compared to reanalysis/observations? I suggest using consistent y-axis ranges for these figures to avoid misleading interpretations, but I understand that the authors may want to compare seasonality patterns rather than magnitudes.

We agree with Reviewer 2 that one y-axis is the most ideal plot layout. Figures 3,4, and 7 have been updated by updating the conversion factor used.

4. Line 110: The gauge record was divided into pre- and post-modification periods. Please provide references to support the chosen separation year. Also, I suggest a more thorough discussion of the influences of human modification on river discharge in the MRB. The main stem of the river was heavily controlled by dams and flood protection structures. The influence of these structures on river flows should be mentioned and highlighted.

We agree with Reviewer 2 that the delineation between pre- and post- modification years needs to be well supported. The years listed in Table 2 are the end of the periods of major river modification when the impacts of the modifications are fully implemented, based on the literature cited in the Table 2 caption. We added text to further clarify the choice of

these years for the subbasins in Section 2.2 (lines 148-179 in the track-changes file; lines 122-129 in the final document).

Minor comments:

1. Line 150: Please provide more details about the lag correlation and spectral angle methods. What formulas were used to calculate these two metrics?

   We agree with Reviewer 2 that this should be specified, and this was updated in the methods section in lines 260-261 in the track-changed file (lines 187-189 of the final document).

2. Table 3: If possible, consider using figures instead of tables to present these results.

   We thank the reviewer for pointing out that clarification is needed. The results originally in Table 3 summarized values from Figures 2,3, and 4 that are discussed in the text. We separated Table 3 into Tables 3, 4, and 5, and placed each below Figures 2, 3, and 4 respectively. We also added figure numbers in the text of Section 3.2 *Other Hydrologic Variables* where they were previously missing.

3. Figure 2: Adding the names of gauges used for each sub-basin would be helpful.

   We agree with Reviewer 2 that this is helpful, and the gage names were added in the caption of Figure 2.

4. Figure 7: Please specify the uncertainty range in the figure caption. Is it a 95% uncertainty interval? Also, clarify the source of uncertainty. Is it coming from inter-annual variability or ensemble variability?

   We agree with the reviewer that this should be specified and it was updated in Figures 2, 3, 4, 7 and Appendix Figures 1, 2, 3, and 4.

---

## Author Response (AR2)

***Editor Decision:***

*We have now received the new reports from both referees. They both agree that the paper has improved substantially and that most comments have been appropriately addressed. However, there are still some concerns mentioned by reviewer #1 about the bias in the results that need to be addressed and/or appropriately rebutted before a final decision.*

**Response:**

We thank  the editor and reviewers and  for their comments and feedback . We believe we have addressed  the comments below , and appreciate hearing back soon.

*Comments from Reviewer 1*

*Most of my concerns have been addressed reasonably. However, although I can see that the reported runoff and snowmelt simulations become more reasonable, the overall CESM1 simulations of river discharge using the RTM routing scheme are still unexpectedly poor. Particularly, the simulations showed the timing bias for the Mississippi River that was not shown in the validation paper of the RTM scheme (Branstetter and Erickson III, 2003). The authors must carefully examine this inconsistency. Also, it would be worth to cross-check with standard CESM1 simulations to ensure that the reported bias is not an error. For example, the authors can verify their results using the ISIMIP2 CLM4.5 simulations which are also based on the RTM routing scheme and follow a well documented protocol.*

*Branstetter, M. L., and D. J. Erickson III (2003), Continental runoff dynamics in the Community Climate System Model 2 (CCSM2) control simulation, J. Geophys. Res., 108, 4550, doi:10.1029/2002JD003212, D17.*

**Response to Reviewer 1**

We thank Reviewer 1 for pointing out the validation in Branstetter and Erickson (2003). We have checked this again and see that our findings are consistent with these other more recent papers, as follows. CESM1, including the Last Millennium Ensemble project, uses the Community Land Model 4.0 (CLM4.0). Branstetter and Erickson III (2003) specifies using the Community Land Model 2.0 (CLM2). As noted, their results show better seasonality, specifically in Figure 1:

[Figure]

However, we find that over the years of different model iterations and in results from others, the seasonality bias is still present. For example, an earlier version from Branstetter (2001) which

uses RTM with the NCAR Land Surface Model 1.0 (LSM1.0) shows model results with peak discharge at nearly the opposite time of year of observations (Figure 2.4g):

**g)**

[Figure]

Brandstetter and Erickson III (2003) is a clear improvement over these early results. In 2015, Li et al investigates the skill of the MOSART and includes RTM in comparisons of skill; both are paired with CLM4.0. In Figure 6, RTM timing, shape, and magnitude are not aligned with observations:

[Figure]

In 2017, Munoz and Dee used CESM1 LME to investigate high flows on the Mississippi. While Supplemental Figure 1 shows overlap in the timing of the largest 10% of events, there again is a mismatch in the distribution where CESM has peak events later and through the year.

[Figure]

**Supplemental Figure 1**. Comparison of lower Mississippi River discharge simulated in the CESM–LME with instrumental data from the gauging station at Vicksburg, Mississippi (USGS station no. 7289000); (a) simulated and instrumental (2008-2015) mean monthly discharge, and month of peak annual discharge for the largest 10% of events in (b) simulated and (c) observed (1897-2015) floods.

Lastly, a previously accessed NCAR CESM1.1 validation resource for "River Flow at Station" for 10 rivers shows a mismatch in timing for the Mississippi. However, the original link (in citations) seems to be temporarily unavailable, likely due to the NCAR migration of CESM data.

[Figure]

Mean Annual Cycle of River Flow at Station

The variation between results shows the need for further validation and explanation of the causes of shifted seasonality in the model. While we discuss in the paper that the MOSART routing model, used in CESM2, is a substantial improvement to RTM, this validation of CESM1 discharge and investigation of causes of shifted seasonality is significant for those who still rely on CESM1 for its Last Millenium Ensemble experiments and to point to the ongoing need for hydrology to be accurately represented in Earth System models.

Citations

Marcia Brandstetter: Development of a Parallel River Transport Algorithm and Applications to Climate Studies, The University of Texas at Austin, 2001.

Branstetter, M. L. and Erickson III, D. J.: Continental runoff dynamics in the Community Climate System Model 2 (CCSM2) control simulation, Journal of Geophysical Research: Atmospheres, 108, https://doi.org/10.1029/2002JD003212, 2003.

Li, H.-Y., Leung, L. R., Getirana, A., Huang, M., Wu, H., Xu, Y., Guo, J., and Voisin, N.: Evaluating Global Streamflow Simulations by a Physically Based Routing Model Coupled with the Community Land Model, Journal of Hydrometeorology, 16, 948–971, https://doi.org/10.1175/JHM-D-14-0079.1, 2015.

Munoz, S. E. and Dee, S. G.: El Niño increases the risk of lower Mississippi River flooding, Scientific Reports, 7, https://doi.org/10.1038/s41598-017-01919-6, 2017.

CESM1.1 Diagnostics: set7_mon_stndisch_10riv.gif (577×912): https://www2.cesm.ucar.edu/experiments/cesm1.1/diagnostics/b.e11.B1850C5CN.f09_g16.001/lnd_1-50-obs/set7/set7_mon_stndisch_10riv.gif, last access: 17 April 2024.

*Comments from Reviewer 2*

*The authors have addressed my comments in this revision. However, I have one concern regarding the use of USGS gauge for the Lower Mississippi River:*
*Figure 1: I cannot agree with using USGS gauge 07295100 Mississippi River at Tarbert Landing to represent the Lower Mississippi River. This is because this gauge is located downstream of the Old River Control Structure (completed in 1963) and heavily influenced by flow regulations (30% of the Mississippi river flow goes into the Atchafalaya River before reaching this gauge). Neither CESM1 nor CESM2 cannot simulate such flow separation in their model settings. I suggest the authors should still use the gauge at Vicksburg in the analysis.*

**Response to Reviewer 2**

We thank Reviewer 2 for this point about the choice of gages. We have updated Figure 2 again, and it now includes both the USGS Mississippi at Vicksburg (07289000) and USACE Mississippi at Tarbert Landing (01100Q), which shows that the seasonality remains an issue between CESM1 and both gages, while also showing the difference in magnitude of discharge. We have also updated Tables 1 and 2 to reflect the change.